# Convergent activation of the integrated stress response and ER–mitochondria uncoupling in VAPB-associated ALS

Curran Landry[1], James P Costanzo[1], Miguel Mitne-Neto[2,3], Mayana Zatz[4], Ashleigh E Schaffer[1,5,8], Maria Hatzoglou [1], Alysson R Muotri [6] & Helen C Miranda [1,5,7 ✉]

## Abstract

Vesicle-associated membrane protein-associated protein-B (VAPB) is an endoplasmic reticulum (ER) membrane-bound protein. The P56S mutation in VAPB causes a dominant, familial form of amyotrophic lateral sclerosis (ALS). However, the mechanism by which this mutation leads to motor neuron (MN) degeneration remains unclear. Utilizing inducible pluripotent stem cell (iPSC)-derived MNs expressing either wild-type (WT) or P56S VAPB, we demonstrate that the mutant protein reduces neuronal firing and disrupts ER-mitochondria-associated membranes (ER MAMs), with a time-dependent decline in mitochondrial membrane potential (MMP), hallmarks of MN pathology. These findings were validated in patient-derived iPSC-MNs. Additionally, VAPB P56S MNs show increased susceptibility to ER stress, elevated expression of the Integrated Stress Response (ISR) regulator ATF4 under stress, and reduced global protein synthesis. Notably, pharmacological ISR inhibition using ISRIB rescued ALS-associated phenotypes in both VAPB P56S and patient-derived iPSC-MNs. We present the first evidence that the VAPB P56S mutation activates ISR signaling via mitochondrial dysfunction in human MNs. These findings support ISR modulation as a strategy for ALS intervention and highlight the need for patient stratification in clinical trials.

**Keywords** ALS (Amyotrophic Lateral Sclerosis); VAPB ((Vesicle Associated Membrane Protein Associated Protein B); ISR (Integrated Stress Response); ER-MAM (Endoplasmic Reticulum Mitochondria Associated Membrane); Neurodegeneration
**Subject Category** Neuroscience

## Introduction

Amyotrophic lateral sclerosis (ALS), also known as Lou Gehrig's disease, is the most common adult–onset motor neuron disease. There is currently no cure for this devastating disorder that affects approximately 30,000 individuals at any time in the United States (Hardiman et al, 2011). A novel, autosomal dominant form of ALS was mapped to chromosome 20q13.3 in a Brazilian family in 2004 (Nishimura et al, 2004). Subsequently, the mutation (NM_004738.5:c.166 C > T, p.(Pro56Ser); hereafter referred to as P56S) in the vesicle-associated membrane protein (VAMP) associated protein *B* (*VAPB*) gene was identified (Mitne-Neto et al, 2007; Nishimura et al, 2004). VAPB contains three different domains: the transmembrane domain, the coiled-coiled domain, and the major sperm protein (MSP) domain, containing the double phenylalanine in an acidic tract (FFAT) binding motif. VAPB is an endoplasmic reticulum (ER) membrane-bound protein that tethers binding proteins and organelles to the ER (Borgese et al, 2021a; Kirmiz et al, 2018). The MSP is the domain known for its protein binding ability and is highly conserved across multiple species, including humans, *D. melanogaster*, *S. cerevisiae* (Borgese et al, 2021b). Lastly, the P56S mutation, located within the MSP domain, impairs VAPB's tethering function (Borgese et al, 2021a; Lev et al, 2008; Mitne-Neto et al, 2011; Mitne-Neto et al, 2007).

One of the main organelles that VAPB anchors to the ER is the mitochondria, and this binding is disrupted by the VAPB P56S mutation. VAPB has been shown to tether the mitochondria to the ER through binding of the mitochondrial tethering protein, protein tyrosine phosphatase interacting protein-51 (PTPIP51) (Obara et al, 2024; Stoica et al, 2014). It has also been shown that this interaction can be present at synapses, and disruptions to it may affect synaptic activity (Gómez-Suaga et al, 2019). VAPB WT interactome studies have been previously performed to help elucidate VAPB functions; however, they have not been carried out in a disease context, in comparison to mutant VAPB P56S, or in motor neurons, the main cell type affected in ALS (Cabukusta et al, 2020; James et al, 2019). Furthermore, the P56S mutation disrupts VAPB normal binding to FFAT (double phenylalanine in an acidic

[1]Department of Genetics and Genome Sciences, Case Western Reserve University, Cleveland, OH, USA. [2]Paulo Gontijo Institute (IPG), São Paulo, Brazil. [3]DNA MM, Sao Paul, SP, Brazil. [4]Human Genome and Stem Cell Center, Biosciences Institute, University of São Paulo, São Paulo, Brazil. [5]Center for RNA Science and Therapeutics, Case Western Reserve University, Cleveland, OH, USA. [6]Department of Pediatrics/Rady Children's Hospital San Diego, School of Medicine, University of California, San Diego, La Jolla, CA, USA. [7]Department of Neurosciences, Case Western Reserve University, Cleveland, OH, USA. [8]Present address: Department of Molecular and Medical Genetics, Oregon Health & Science University, Portland, OR, USA. ✉E-mail: hcm34@case.edu

tract) motif containing proteins and is also known to cause a conformational change, leading VAPB proteins to aggregate. Protein aggregates within cells expressing both wild-type VAPB (VAPB WT) and VAPB P56S have been shown to include VAPB WT into the aggregates, demonstrating sequestration of VAPB WT as another possible mechanism for pathogenesis (Kanekura et al, 2006; Teuling et al, 2007). Interestingly, the only previously published VAPB iPSC-derived disease model showed that ALS type VIII patient iPSC-derived neurons have decreased levels of soluble VAPB (Mitne-Neto et al, 2011).

To date, there is no effective treatment for ALS, and current approved therapies only extend lifespan for a few months, partially due to a lack of understanding of disease pathogenesis. Recent studies have suggested that activation of the integrated stress response (ISR) pathway could be a link between forms of ALS, as well as a possible therapeutic target for this disease (Bugallo et al, 2020; Marlin et al, 2022). The ISR is a conserved intracellular signaling network that is activated in response to a range of stressors, with a canonical convergence in the phosphorylation of the eukaryotic initiation factor 2 alpha (eIF2α) (Brostrom et al, 1996; Harding et al, 2003; Wek et al, 2006). Once phosphorylated, p-eIF2α then has two major canonical effects, the inhibition of cap-dependent translation as well as the specific upregulation of the expression of effector translation factors, namely activating transcription factor 4 (ATF-4) to exert the cell's response to the initiating stress (Dever et al, 1992; Harding et al, 2003; Pakos-Zebrucka et al, 2016). Two Phase I clinical trials targeting the integrated stress response (ISR)—DNL343 and ABBV-CLS-7262—were recently completed, but neither achieved their primary endpoints related to motor function or survival. These outcomes underscore the urgent need for mechanistic insight into how and when ISR modulation may benefit ALS patients. Emerging evidence suggests that ISR involvement may differ across genetic subtypes of ALS or represent a downstream consequence of other cellular dysfunctions. Although ISR activation has been observed in various ALS models, a mutation-specific mechanistic connection has remained elusive (Chen et al, 2021; Marlin et al, 2022; Mejzini et al, 2019). Here, we define such a connection by demonstrating that the VAPB P56S mutation—associated with ALS8—elicits ISR activation via mitochondrial dysfunction in human motor neurons. These findings establish a mutation-driven cascade linking ER–mitochondria uncoupling to ISR dysregulation, offering a novel framework for developing targeted and stratified therapeutic approaches.

In this study, we generated iPSCs stably expressing either the VAPB P56S or VAPB WT in a CRISPR-Cas9 knockout VAPB background to interrogate VAPB P56S-specific pathogenesis associated with ALS. We differentiated the iPSCs into motor neurons and identified an electrophysiological activity dysregulation by a multi-electrode array (MEA) system. We then explored the molecular mechanism of this dysregulation and found that VAPB P56S has reduced binding with many mitochondrial binding partners, including PTPIP51 (Gomez-Suaga et al, 2017; Stoica et al, 2014). Given this, we examined the ER mitochondrial-associated membrane (ER MAM) and found that the VAPB P56S motor neurons displayed decreased ER MAM compared to VAPB WT throughout differentiation. Moreover, we identified a culture time-dependent decline in mitochondrial membrane potential (MMP). Importantly, we observed that the VAPB P56S iPSC-derived motor neurons were more sensitive to cell stressors, exhibiting early activation of the ISR, namely through the DELE1-mediated pathway triggered by mitochondrial stress. We also generated an iPSC line from an ALS patient harboring the VAPB P56S mutation and an isogenic control through CRISPR-Cas9 technology, and we were able to validate these findings in the ALS-iPSC-derived motor neurons. Using an ISR inhibitor (ISRIB), we were able to rescue the neuronal firing and MMP dysfunctions, indicating that they were manifestations of ISR activation. To our knowledge, this represents the first comprehensive study of VAPB P56S in ALS-relevant cells —iPSC-derived motor neurons—sharing the same genetic background, providing the most thorough mechanistic investigation of ALS type VIII pathogenesis to date. This study also provides the first mechanistic link between the ISR and the ALS-associated variant, VAPB P56S.

# Results

## Decreased electrophysiological activity in VAPB P56S motor neurons

To investigate the effect of the VAPB P56S mutant independently from VAPB WT, as well as the mutation's effect on binding properties of VAPB protein and disease phenotype more broadly, we created isogenic iPSC lines stably expressing either VAPB WT or VAPB P56S. First, we knocked out endogenous VAPB using CRISPR-Cas9 technology and then generated lentiviral expression vectors containing the mutant or the wild-type genes under the control of a tetracycline-dependent promoter (Fig. EV1A–D). Once stably transduced and selected, VAPB expression is controlled in a dose-dependent manner with the addition of doxycycline. A doxycycline dose-response curve for exogenous VAPB expression in the transduced iPSCs was performed in comparison to endogenous VAPB expression in patient iPSC harboring the VAPB P56S mutation and familial unaffected controls (VAPB WT) to select the physiological doxycycline dose for both VAPB WT and VAPB P56S (0.2 ug/mL) (Fig. EV2A). All further experiments were performed in iPSC-derived motor neurons, differentiated using a dual SMAD inhibition protocol, as previously described (Fig. 1A) (Markmiller et al, 2018). The cultures predominantly contain mature motor neurons (MN) on day 25. iPSCs were treated with doxycycline to express either VAPB WT or VAPB P56S from day 1 of differentiation into motor neurons. No significant difference was noted between the VAPB WT and VAPB P56S doxycycline-inducible lines, in either efficiency of motor neuron progenitor (Fig. EV2B) or motor neuron differentiation (Fig. EV2C).

To examine whether the VAPB P56S mutation affects neuronal activity, a relevant ALS phenotype, we used an MEA system. Mature VAPB P56S iPSC-derived motor neuron recordings were taken every other day and compared to VAPB WT iPSC-derived motor neurons. Just after day 40, the iPSC-derived motor neuron cultures reached a plateau in electrophysiological activity. The VAPB P56S iPSC-derived motor neurons show significantly decreased firing from day 40 onward when compared to VAPB WT control (Figs. 1B and EV3A).

Additionally, we quantified the neuronal bursting of our iPSC-derived motor neurons to assess whether the P56S mutation was affecting the firing rate independently of bursting or showing a

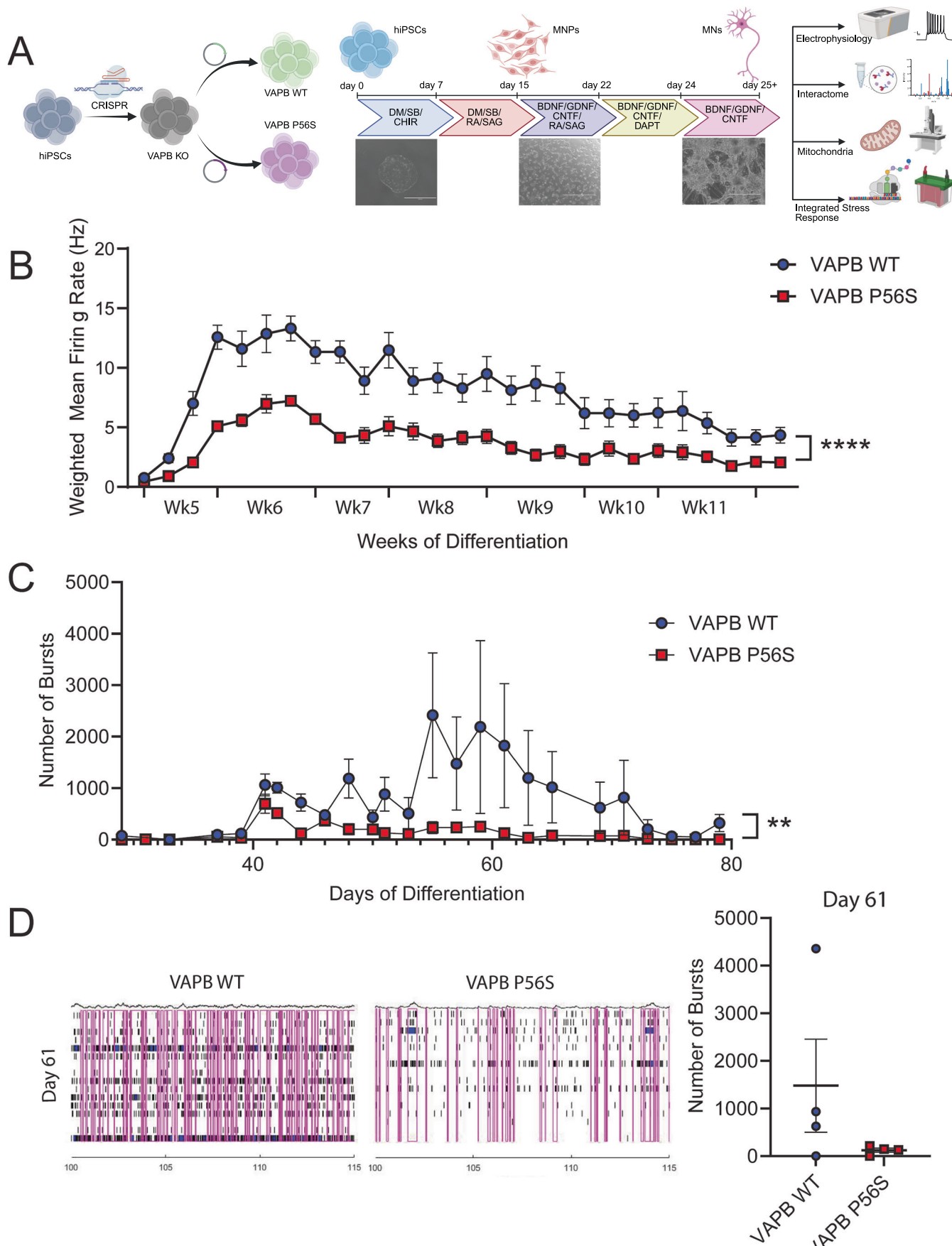

◄ **Figure 1. VAPB P56S motor neurons exhibit decreased neuronal firing rate compared to WT controls.**

(A) Schematic representation of the motor neuron differentiation timeline. Created in BioRender. (https://BioRender.com/6g7xwxj and https://BioRender.com/6drsh7). (B) Weighted mean firing rate throughout 11 weeks of motor neuron differentiation. Data were presented as mean ± SEM, $N = 8$ technical replicates from each of three biological replicates. Statistical analysis: mixed-effects model (REML), interaction effect $p = 0.000006$****. (C) Number of bursts recorded through day 80 of motor neuron differentiation. Data were presented as mean ± SEM, $N = 8$ technical replicates. Statistical analysis: two-way ANOVA, interaction effect $p = 0.0054$**. (D) Raster plot depicting neuronal firing on day 61, recorded 100 s post-start. Bursts are outlined in magenta. Scatter plot quantifying the total number of bursts per well on day 61 of differentiation. Data were presented as mean ± SEM, $N = 4$ replicates. Source data are available online for this figure.

commensurate decrease in the network firing of the VAPB P56S iPSC-derived motor neurons compared to VAPB WT controls. We found that only the VAPB WT iPSC-derived motor neurons show consistently increased bursting after day 55 (Fig. 1C). We also examined histogram and scatter plots of the neuronal activity later in differentiation (day 61) to assess the specific firing of the motor neurons subjected to extended time in culture, as ALS is neurodegenerative disease (Fig. 1D). We observed that the VAPB WT iPSC-derived motor neurons had increased bursting in addition to their increased firing rate compared to the VAPB P56S iPSC-derived motor neurons, indicating greater connectivity and synchronicity within the VAPB WT iPSC-derived motor neuron cultures.

## VAPB P56S leads to perturbation of mitochondrial binding partners

VAPB is a tethering protein; therefore, in order to elucidate the molecular pathogenesis, we chose to analyze the binding partners of VAPB and how the P56S mutation affects them. We purposely developed the VAPB inducible expression model system to allow for the isolation of VAPB P56S binding partners and thus define the VAPB P56S interactome independently of VAPB WT, unachievable using patient lines, as patients are heterozygous for the P56S mutation. To this end, we cultured iPSC-derived motor neurons with inducible VAPB WT or VAPB P56S, as well as a VAPB KO line to be used as a negative control. Protein isolation was performed at day 35 of the motor neuron differentiation protocol. VAPB and associated proteins were immunoprecipitated (IP) and then submitted for mass spectrometry-based peptide identification (Fig. 2A).

We compared the VAPB WT and VAPB P56S-associated proteins to identify bindings that were disrupted in the VAPB P56S interactome. The list of differentially bound interactors was then analyzed in the ingenuity pathway analysis (IPA) software to identify pathways in which the binding of VAPB would be most disrupted under disease conditions (Fig. 2B). We then had IPA generate connections between the most highly enriched pathways and found two clusters, one consisting of metabolic pathways and the other containing cellular processes. The only connection between these two clusters was the pathway of mitochondrial dysfunction (Fig. 2C). Given the possible significance of the mitochondrial dysfunction pathway identified in IPA, we also used the Database for Annotation, Visualization and Integrated Discovery (DAVID) to identify cellular compartments enriched for proteins with loss of binding with VAPB P56S compared to VAPB WT (Fig. 2D). The compartment with the second highest number of interactors lost is the mitochondria, succeeding only the cytoplasm, where it is expected that the majority of the soluble

proteins reside. The results from both IPA and DAVID implicated mitochondrial binding partners as the disrupted interaction in VAPB P56S expressing iPSC-derived motor neurons corroborating previously published data showing that VAPB is a critical component in tethering the mitochondria to the ER, through the interaction with PTPIP51 (De Vos et al, 2012; Gomez-Suaga et al, 2017; Obara et al, 2024; Stoica et al, 2014). De Vos et al showed that the P56S mutation actually increases VAPB's affinity for PTPIP51, and while we corroborated these findings, we also found that given the decrease in soluble VAPB found not only in our model system (Fig. EV2A), but in other patient-relevant systems, the total amount of PTPIP51 pulled down in the VAPB IP from the VAPB P56S iPSC-derived motor neurons is significantly decreased compared to that of the VAPB WT iPSC-derived motor neurons (Figs. 2E and EV4) (Anagnostou et al, 2010; Borgese et al, 2021a; Cadoni et al, 2020; De Vos et al, 2012; Suzuki et al, 2009; Teuling et al, 2007). Thus, our results are in agreement with previous studies showing that while the P56S mutation increases VAPB's affinity for PTPIP51, due to the reduction in soluble VAPB levels, there is a reduction in this interaction within the affected cells, and we identified this phenomenon is present in motor neuron cells with untagged VAPB WT or P56S (De Vos et al, 2012; Obara et al, 2024; Stoica et al, 2014).

## Compromised ER-mitochondrial contact sites and mitochondrial dysfunction in VAPB P56S motor neurons

To investigate if the loss of VABP P56S association to proteins in the mitochondria could alter the ER MAM, we performed electron microscopy analysis to quantify the extent of membrane contact between the ER and the mitochondria. We analyzed day 35 and day 60 VAPB WT or VAPB P56S expressing iPSC-derived motor neurons, to examine not only if VAPB P56S affects ER MAM, but whether prolonging the iPSC-derived motor neurons time in culture may alter ER MAM, since we observed decreased neuronal firing after day 40 of differentiation (Fig. 1B). We found that the VAPB P56S iPSC-derived motor neurons exhibited a significant decrease in the percentage of mitochondrial contact with the ER as early as day 35 (Fig. 3A) which persisted on day 60 (Fig. 3B) of differentiation. (Figs. EV5A–C and EV6A,B). ER MAMs are crucial sites for maintaining proper mitochondrial bioenergetics; therefore, we next had to assess the consequence of this loss of ER MAM on mitochondrial function (van Vliet and Agostinis, 2018). To investigate this, we used the JC-1 assay to measure the mitochondrial membrane potential (MMP). Given the VAPB P56S iPSC-derived motor neurons decrease in electrophysiological firing after day 40, we decided to assess MMP in iPSC-derived motor neurons at two timepoints, shortly after neuronal maturation, day 30, and after extended time in culture, day 60.

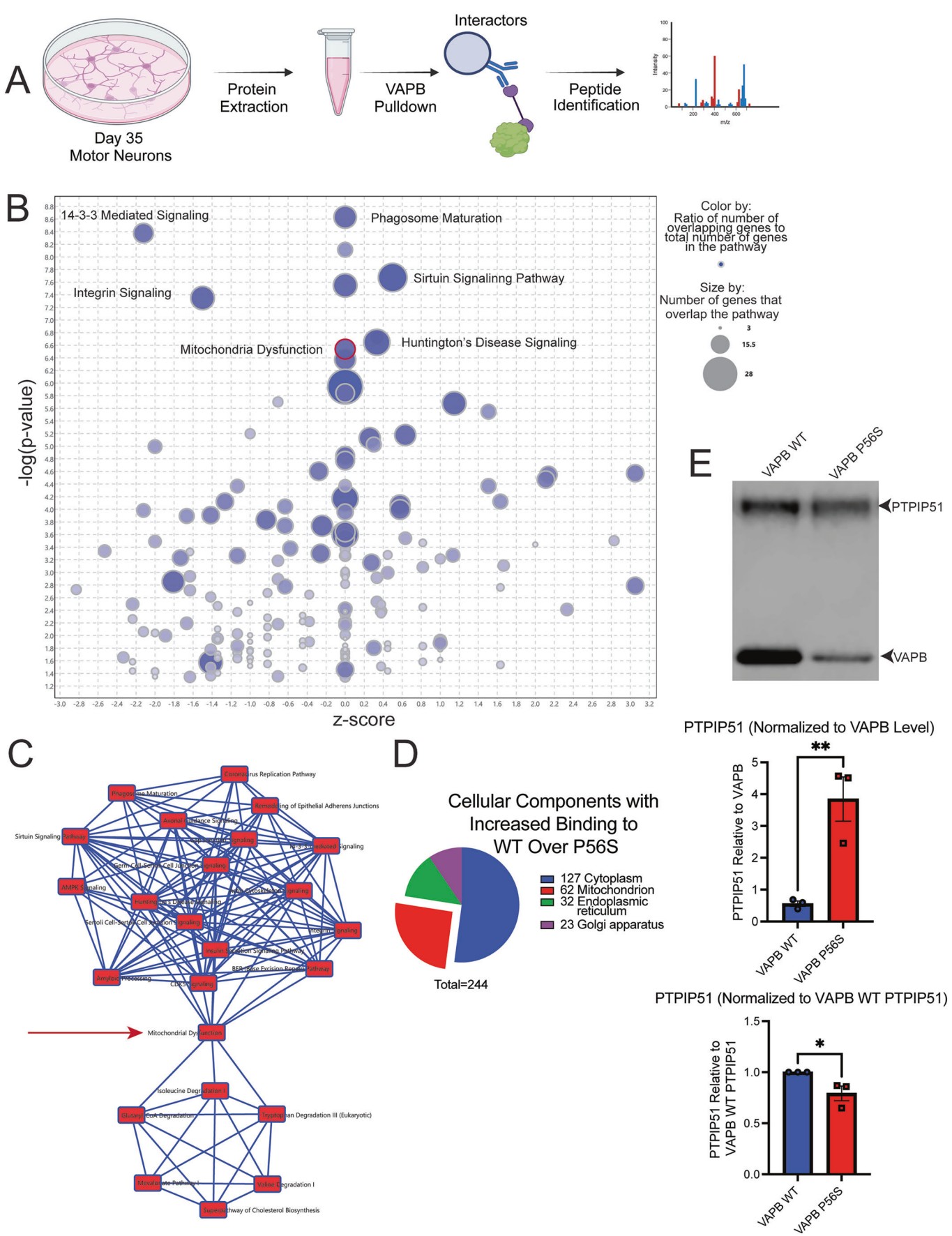

◀ **Figure 2. Interactome analysis of VAPB P56S compared to VAPB WT.**

(A) Schematic illustrating the co-immunoprecipitation workflow used to isolate and identify VAPB interactors. (B) Ingenuity pathway analysis (IPA) depicting pathways enriched in proteins that exhibit increased binding to WT over P56S. The X-axis represents the IPA z-score, indicating predicted pathway activation/inhibition, while the Y-axis represents the −log($p$ value) for pathway enrichment significance. (C) A diagram showing connections between highly enriched pathways identified via IPA. (D) Cellular compartment analysis of proteins with a ≥1.5-fold increase in binding to VAPB WT compared to VAPB P56S, analyzed using DAVID. (E) Western blot of VAPB immunoprecipitation samples stained for PTPIP51 and β-actin. Quantification normalized to corresponding VAPB levels, mean ± SEM, $N = 3$ biological replicates. Statistical analysis: unpaired $t$-test, $p = 0.0093$**. Additional quantification of PTPIP51 levels in VAPB P56S samples normalized to VAPB WT levels, mean ± SEM, $N = 3$ biological replicates. Statistical analysis: unpaired $t$-test, $p = 0.0439$*. Source data are available online for this figure.

Interestingly, we observed that the day 60 VAPB P56S iPSC-derived motor neurons had a significantly increased green-fluorescing population, indicating decreased MMP compared to the VAPB WT controls, whereas on day 30, there was no significant difference between VAPB WT and VAPB P56S iPSC-derived motor neurons (Fig. 3C,D). These findings suggest that although the reduction in ER MAM is reduced throughout the lifespan of the VAPB P56S iPSC-derived motor neurons, the functional impact on the MMP requires time to manifest.

## VAPB P56S impairs the cellular ability to adapt to cell stressors

ER MAMs are known to be a hot spot for the transfer of stress signals from the ER to mitochondria (van Vliet and Agostinis, 2018). Consequently, to test whether we could induce mitochondrial dysfunction by exposing the iPSC-derived motor neurons to stressors, we selected tunicamycin (TM), an ER stressor, as VAPB is an ER protein, and if the decreased MMP could be caused, at least partially, by loss of ER MAM, ER stress would likely exacerbate it. We first treated iPSC-derived motor neurons on day 35 of differentiation with TM, for either 24 or 48 h, and then assayed MMP. There was no change in the MMP at the 24 h time point, however, at 48 h of treatment, the VAPB P56S iPSC-derived motor neurons display a significant decrease in MMP compared to the VAPB WT (Fig. EV7A). This indicates that VAPB P56S iPSC-derived motor neurons are more sensitive to stress compared to VAPB WT, with VAPB P56S iPSC-derived motor neurons exhibiting a significantly lower MMP compared with VAPB WT after 48 h of TM treatment.

Interestingly, when the TM treatment was performed on day 60 of the iPSC-derived motor neuron differentiation, both VAPB WT and VAPB P56S iPSC-derived motor neurons have decreased MMP after 24 h; however, after 48 h, the VAPB WT motor neurons were able to spontaneously adapt to the TM stress, as shown by the increase in their MMP, and are no longer significantly different than the VAPB WT vehicle control. In contrast, the VAPB P56S iPSC-derived motor neurons show persistent decreased MMP at the 48 h mark, indicating a failure to adapt to the TM challenge (Fig. EV7B). Taken together, these data suggest that not only is the physical disruption of the ER MAM underlying the functional deficits observed later in differentiation, but also suggest that the VAPB P56S iPSC-derived motor neurons have an increased sensitivity to stress, and an impaired adaptation to it.

## VAPB P56S triggers mutation-specific activation of the integrated stress response via mitochondrial stress

Induction of ER stress is known to activate the integrated stress response (ISR), an intracellular mechanism that helps the cell to adapt to a diverse set of stimuli and maintain homeostasis through downregulation of global mRNA translation and upregulation of specific genes, namely ATF-4 (B'chir et al, 2013; Kilberg et al, 2009; Pakos-Zebrucka et al, 2016). We performed western blotting of ATF-4 from whole protein extract of VAPB WT or VAPB P56S iPSC-derived motor neurons on day 35 and day 60 of differentiation. We observed no basal protein expression of ATF-4 in either VAPB WT or VAPB P56S iPSC-derived motor neurons on day 35, and low-level expression on day 60, with no significant difference between VAPB WT and VAPB P56S. After TM treatment, both lines display statistically significant increases in ATF-4 expression, as expected in the context of ISR activation. However, in a comparison between VAPB P56S and VAPB WT lines in the context of TM treatment, in both day 35 and day 60 iPSC-derived motor neurons, the VAPB P56S expresses a significantly higher level of ATF-4 than the VAPB WT (Fig. 4A,B).

Once the increased sensitivity to stress of the VAPB P56S iPSC-derived motor neurons was established, we wanted to elucidate the molecular link between ER and mitochondrial dysfunction. Previous work shows that mitochondrial stress leads to DELE1 cleavage, and once cleaved, DELE1-s then binds heme-regulated eIF2α kinase (HRI), activating it, leading to the phosphorylation of eIF2α (Bauer et al, 2001; Fessler et al, 2020; Guo et al, 2020; Rafie-Kolpin et al, 2003). Once phosphorylated, p-eIF2α is known to cause expression of ATF-4, a key regulator in the integrated stress response pathway (ISR). In addition, the ISR is a key cellular process that is known to be linked to ER and mitochondrial health (Chen et al, 2021; Costa-Mattioli and Walter, 2020; Mick et al, 2020). Given this and our data showing disruptions in ER MAM, mitochondrial function, and ATF-4 upregulation in response to a stressor, we decided to examine the ISR in the VAPB P56S iPSC-derived motor neurons. We then assessed DELE1 as mitochondrial stress causes cleavage of DELE1 (also known as DELE1-L) into small (DELE1-s) fragments (Bauer et al, 2001; Fessler et al, 2020; Guo et al, 2020; Rafie-Kolpin et al, 2003). When we performed western blotting for DELE1 from whole protein extract of VAPB WT or VAPB P56S iPSC-derived motor neurons, the levels of DELE1 we found in day 35 VAPB P56S iPSC-derived motor neurons display increased levels of DELE1-s compared to VAPB WT controls, with no difference in levels of DELE1-L (Fig. 4C). Furthermore, we analyzed p-eIF2α as the downstream effector of the increase in DELE1-s and saw that the day 35 VAPB P56S iPSC-derived motor neurons exhibit a higher ratio of p-eIF2α:eIF2α when compared to VAPB WT counterparts (Fig. 4D).

In addition to activating ATF-4, p-eIF2α also suppresses global mRNA translation; therefore, we decided to assess the level of protein synthesis in day 35 motor neurons (Baird and Wek, 2012;

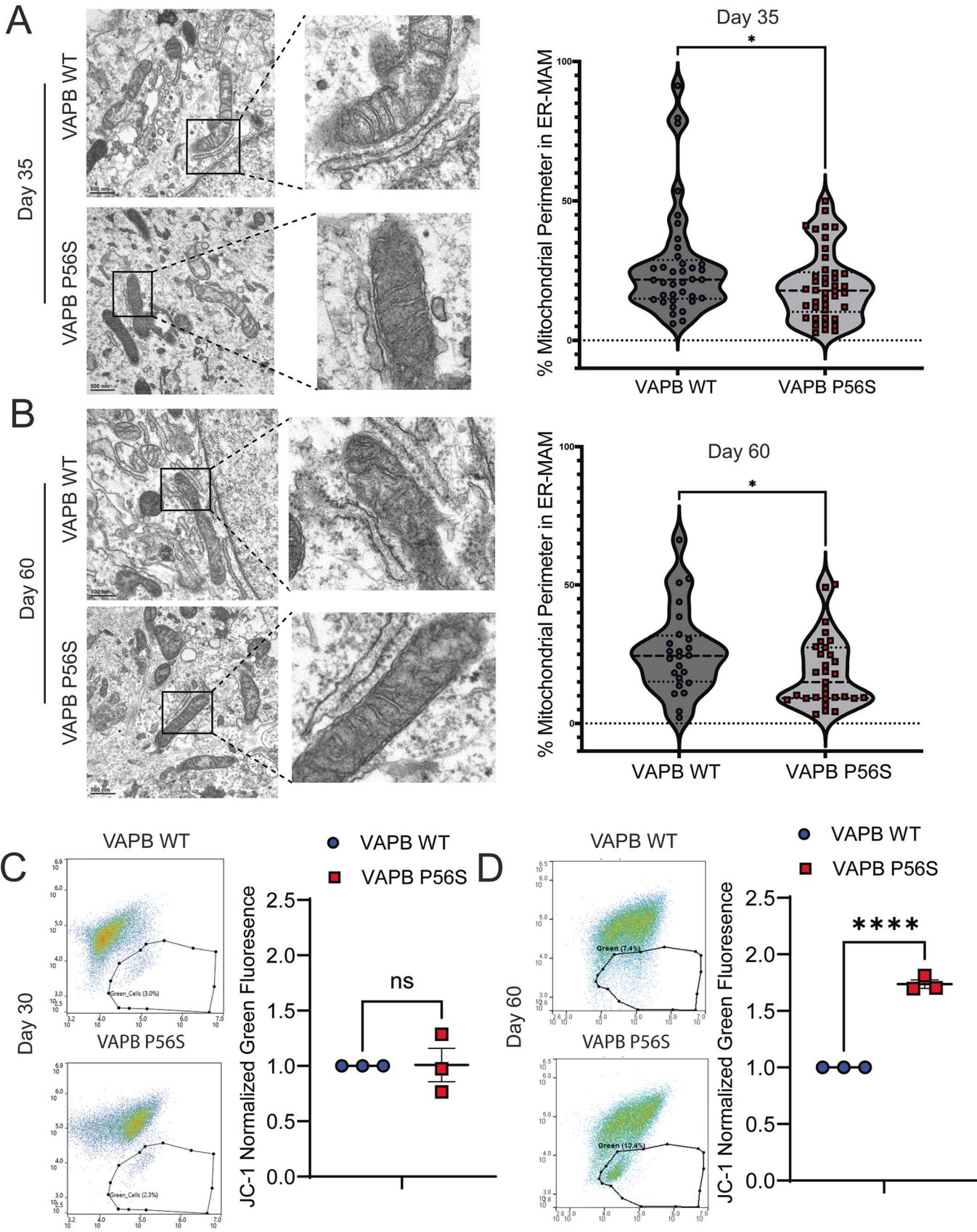

**Figure 3. VAPB P56S reduces ER-mitochondrial contact and decreases mitochondrial membrane potential.**

(A) Electron microscopy (EM) images of day 35 VAPB WT and VAPB P56S motor neurons, with inserts highlighting mitochondria-ER contacts. Violin plot represents median, first, and third quartiles (dashed lines), $N = 37$ mitochondria from 13 unique images for D35 WT, 42 mitochondria from 21 unique images for D35 P56S, unpaired $t$-test, $p = 0.0475*$. (B) EM images of day 60 VAPB WT and VAPB P56S motor neurons, with inserts highlighting mitochondria-ER contacts. Violin plot represents median, first, and third quartiles (dashed lines), $N = 24$ mitochondria from 13 unique images for D60 WT, 31 mitochondria for 18 unique images for D60 P56S, unpaired $t$-test, $p = 0.0485*$. (C) Scatter plot of day 30 motor neurons stained with JC-1 mitochondrial dye. X-axis: red fluorescence; Y-axis: green fluorescence; gate indicates the highly green-fluorescing population. Data presented as mean ± SEM, $N = 3$ biological replicates, paired $t$-test, $p > 0.05$. (D) Scatter plot of day 60 motor neurons stained with JC-1 mitochondrial dye. X-axis: red fluorescence; Y-axis: green fluorescence; gate indicates the highly green-fluorescing population. Data presented as mean ± SEM, $N = 3$ biological replicates, paired $t$-test, $p = 0.0025**$. Source data are available online for this figure.

Harding et al, 2000). To examine this, we utilized the surface sensing of translation (SUnSET) assay to ascertain the level of translation through the rate of puromycin incorporation (Goodman and Hornberger, 2013; Schmidt et al, 2009). We observed a significant decrease in the amount of puromycin incorporation in the VAPB P56S iPSC-derived motor neurons, signaling a decrease in the level of translation on day 35, consistent with the elevated ratio of p-eIF2α:eIF2α (Fig. 4E). Therefore, we can surmise that the activation of ISR is increased in the VAPB P56S over the VAPB WT iPSC-derived motor neurons, through the mitochondrial stress sensing pathway of DELE1.

### Targeting the ISR reverses VAPB P56S-induced ALS phenotypes: therapeutic implications

After establishing that the VAPB P56S mutation triggers ISR activation in iPSC-derived motor neurons, we aimed to determine whether the ISR was at least partially responsible for the observed functional phenotypes. To this end, we treated iPSC-derived motor neuron cultures with integrated stress response inhibitor (ISRIB), which, when bound to eIF2B (a key component of a translational initiation complex) induces a conformational change in eIF2B prompting dissociation of p-eIF2α thereby promoting translation, inhibiting ATF-4 expression, and blunting the ISR (Zyryanova et al, 2021). We first interrogated ISRIB's effect on protein synthesis using the SUnSET assay on day 35 and observed that ISRIB treatment significantly increased puromycin incorporation in VAPB P56S iPSC-derived motor neurons. Protein synthesis levels in ISRIB-treated VAPB WT and VAPB P56S were comparable, indicating that ISRIB treatment restored decreased protein synthesis caused by VAPB P56S-induced ISR (Fig. 5A). As expected, protein synthesis did not significantly change in VAPB WT expressing iPSC-derived motor neurons subjected to ISRIB, likely due to limited ISR activation in these cells. We wanted to next examine the effect of ISRIB on the decreased MMP observed in VAPB P56S expressing iPSC-derived motor neurons. After treatment with ISRIB for 24 h on day 60, both VAPB WT and VAPB P56S iPSC-derived motor neurons showed a significant increase in MMP, with no significant difference between the two lines after treatment, indicating that ISRIB ameliorates the decreased MMP phenotype, and that the increased ISR activity in VAPB P56S iPSC-derived motor neurons likely underlies the decreased MMP (Fig. 5B). Finally, we wanted to examine whether the decreased neuronal firing observed in VAPB P56S iPSC-derived motor neurons could be rescued through ISRIB treatment. Therefore, we began ISRIB treatment on day 45 of differentiation and observed a significant increase in the firing rate of VAPB P56S

iPSC-derived motor neurons treated with ISRIB compared to that of vehicle-treated VAPB P56S iPSC-derived motor neurons (Fig. 5C). Along with decreased MMP and translation, the reduction in neuronal firing was at least partially caused by an activation of the ISR within the VAPB P56S iPSC-derived motor neurons.

### Restoration of mitochondrial function and protein synthesis in ALS type VIII patient-derived motor neurons

We synthesized the doxycycline-inducible VAPB lines described in the experiments above so as to examine the effects of the mutant VAPB P56S without the interference of the wild-type protein, as patients with the P56S mutation are heterozygous. While the doxycycline-inducible lines provided clearer insight into the mechanism by which the VAPB P56S mutation leads to motor neuron degeneration (Fig. 2), it is also true that this model system may not fully recapitulate the disease pathogenesis within the patients (Fig. EV2A). To determine whether the mechanism we elucidated in the doxycycline-inducible lines was indeed applicable to the patient cellular environment, we used an ALS8 patient fibroblast line, heterozygous for the VAPB P56S mutation, and reprogrammed it in vitro to create an iPSC line. We then used CRISPR-Cas9 to correct the mutation and create a VAPB homozygous wild-type isogenic control (Fig. EV8). We then used this ALS patient-derived iPSC to validate the phenotypes we had previously identified in the doxycycline-inducible VAPB P56S iPSC. First, we assessed global protein synthesis by SUnSET assay on day 35 and validated the decrease in translation observed in the VAPB P56S doxycycline-inducible iPSC-derived motor neurons, which was also rescued upon ISRIB treatment (Fig. 6A). A JC-1 assay was performed on day 35 of differentiation and identified an increase in the green, fluorescent population of the patient iPSC-derived motor neurons, indicating a decrease in MMP, just as was observed in the VAPB P56S doxycycline-inducible line. The motor neurons were then treated with ISRIB and showed that treatment with ISRIB reverses this decrease in the MMP, rescuing the phenotype (Fig. 6B). Taken together, our data suggest that the P56S mutation causes a disruption in VAPB binding to mitochondrial proteins, leading to a decrease in ER MAM. This causes mitochondrial stress, which in turn activates the Integrated Stress Response Pathway through DELE1 cleavage. Activation of the ISR then leads to decreased neuronal firing and eventually decreased MMP in VAPB P56S iPSC-derived motor neurons, which was confirmed in iPSC-derived motor neurons from an ALS patient harboring the VAPB P56S mutation from a different genetic background (Fig. 7).

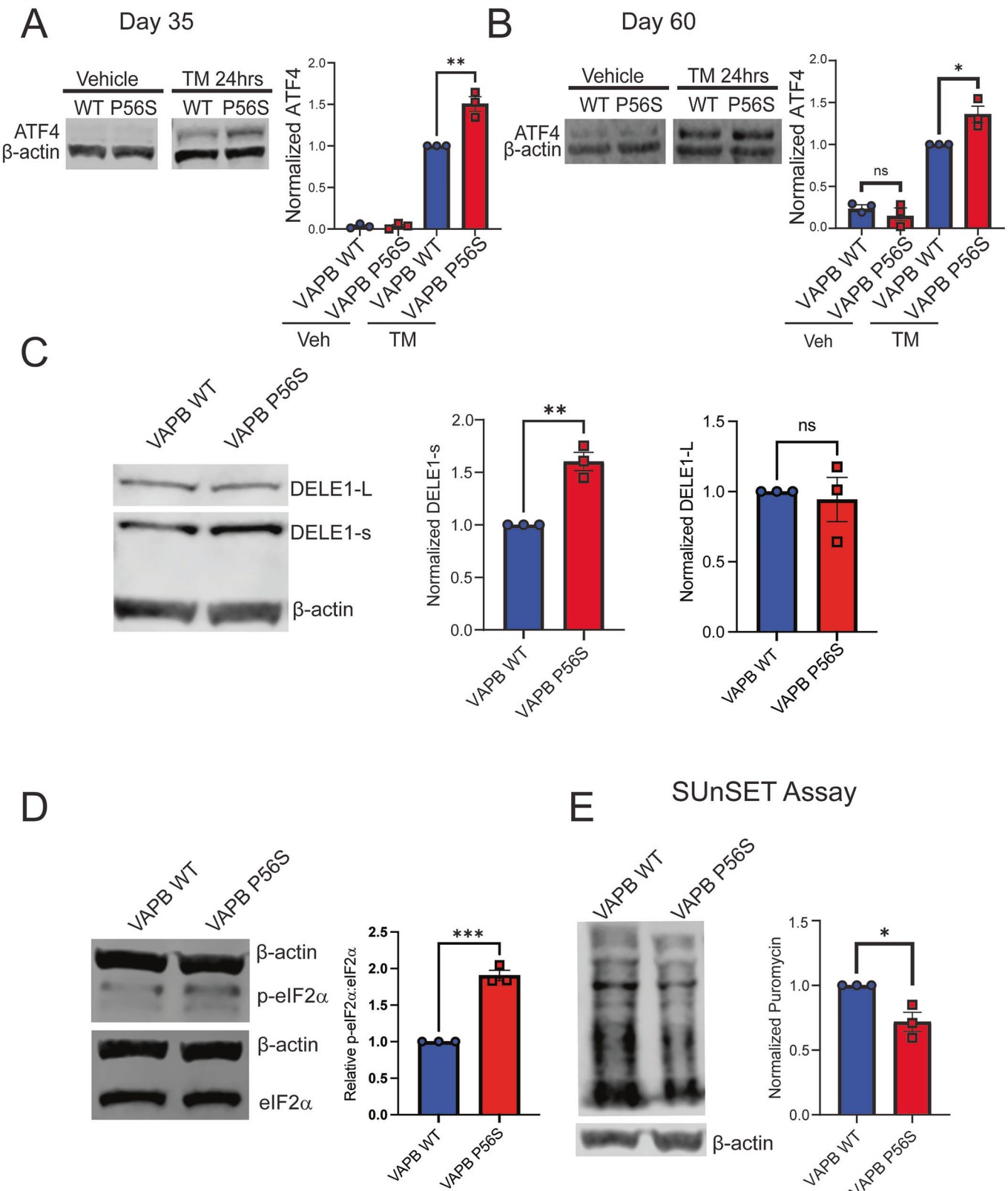

**Figure 4. The integrated stress response (ISR) is activated in VAPB P56S motor neurons.**

(A) Western blot of ATF-4 basal levels and after 24 h tunicamycin exposure in day 35 motor neurons. Data presented as mean ± SEM, $N = 3$ biological replicates, unpaired $t$-test, $p = 0.0045^{**}$. (B) Western blot of ATF-4 basal levels and after 24 h tunicamycin exposure in day 60 motor neurons. Data presented as mean ± SEM, $N = 3$ biological replicates, unpaired $t$-test, $p = 0.0217^*$. (C) Western blot of DELE1 large and small fragments on day 35 of differentiation. Data presented as mean ± SEM, $N = 3$ biological replicates, unpaired $t$-test, $p = 0.0023^{**}$. (D) Western blot of phosphorylated eIF2α (p-eIF2α) and total eIF2α on day 35 of differentiation. Data presented as mean ± SEM, $N = 3$ biological replicates, unpaired $t$-test, $p = 0.0002^{***}$. (E) SUnSET assay on day 35 of differentiation. Data presented as mean ± SEM, $N = 3$ biological replicates, unpaired $t$-test, $p = 0.0188^*$. Source data are available online for this figure.

## Discussion

Despite decades of research, the mechanisms by which specific ALS-causing mutations lead to neurodegeneration remain poorly defined, limiting therapeutic advances. Here, we elucidate a novel pathogenic cascade linking the VAPB P56S mutation to mitochondrial stress, ISR activation, and functional motor neuron deficits in human iPSC-derived models. This is the first study to mechanistically connect a known ALS mutation with ISR activation, highlighting the potential for mutation-specific therapeutic targeting and patient stratification in ISR-modulating clinical trials. Decreased ER MAM and initiation of the ISR precede the reduction of MMP and decreased motor neuron electrophysiological activity. Combined with the fact that challenges with stressors such as tunicamycin early in differentiation replicate late-stage disease-relevant phenotype presentation, and that ISRIB reverses these phenotypes, we infer that the underlying physical deficit of decreased ER MAM initiates ISR activation that potentially intensifies over time, eventually resulting in motor neuron degeneration. Prior investigations have implicated the ISR as a potential molecular mechanism associated with other variants of ALS. A study by Saxena, Sabuy, and Caroni in 2009 showed that ER stress influences disease progression in vulnerable motor neuron populations in SOD1 mice, providing evidence for the link between ER dysfunction and motor neuron degeneration (Saxena et al, 2009). In addition, the reduction in ER MAM through VAPB reduction or mutation has also previously been shown, albeit not in disease-relevant models (Chen et al, 2021; Obara et al, 2024; Stoica et al, 2014). However, a comprehensive understanding of the connection between ALS variants and the ISR pathway has remained elusive until now, as some studies indicate that ISR blunting is protective, while others suggest it is detrimental (Bugallo et al, 2020; Marlin et al, 2022).

Our data support that either decreasing ISR or increasing stress adaptation mechanisms can benefit inhibition of motor neuron disease phenotypes in a time and condition-dependent manner. To our knowledge, this represents the most thorough characterization of VAPB P56S pathogenesis to date, clarifying the role of the ISR and its connection to the mutation. Moreover, we found ISRIB treatment rescues disease-relevant phenotypes caused by the VAPB P56S mutation, including the most clinically relevant, motor neuron electrophysiology. Thus, we suggest that dampening the ISR in motor neurons may be a valid therapeutic approach in patients with the VAPB P56S mutation. As previously mentioned, there were two clinical trials investigating this approach in a non-stratified ALS population; the clinical trials were openly recruiting sporadic and familial ALS patients (ClinicalTrials.gov Identifiers: NCT04948645 and NCT05842941). While they did not meet primary endpoints for survival or movement, we suggest that the data presented here advocate for the stratification of ALS patients, and a more personalized therapeutic approach may be required. Indeed, while some studies have shown a link to ISR in ALS, others have shown that therapeutic targeting of the ISR in SOD1 forms of ALS may, in fact, aggravate disease onset (Marlin et al, 2022). Thus, more work remains to be done to elucidate how common ISR activation is in various forms of ALS pathogenesis, which patients will benefit from such intervention, and whether the therapy would serve as a preventive measure or remain effective after motor neuron degeneration has already started.

Although we have identified a pathway for ISR activation in VAPB P56S cells, the precise mechanism by which ISR activation leads to neurodegeneration remains unclear, presenting another potential target for therapeutic intervention. It may involve translational suppression mediated by p-eIF2α. This could result in a global reduction of protein synthesis or the downregulation of a specific protein essential for neuronal health. Further research is necessary to delineate this process. Identifying the precise pathophysiological mechanisms underlying disease progression will enable the development of targeted interventions aimed at disrupting pathogenic pathways and restoring cellular homeostasis. Previous studies have demonstrated that the VAPB P56S mutation reduces ER-MAMs; however, these studies have not proposed a downstream pathogenic cascade arising from this defect (Gómez-Suaga et al, 2019; Obara et al, 2024; Stoica et al, 2014).

Recent findings underscore the critical role of VAPB-PTPIP51 binding in mitochondrial function, further emphasizing the significance of our results. Shiiba et al (2025) reported that disruption of VAPB-PTPIP51 tethering leads to increased mitochondrial reactive oxygen species (ROS), while Wilson et al (2025) found that VAPB P56S aggregation correlates with mitochondrial dysfunction. Nevertheless, no study has yet established a complete VAPB P56S ALS pathogenesis (Shiiba et al, 2025; Wilson et al, 2025).

We propose that ER-mitochondrial detachment is the most probable factor sensitizing cells to stressors, as it represents a direct consequence of the VAPB P56S mutation. This hypothesis aligns with previous findings and extends to multiple forms of ALS via diverse mechanisms, both direct and indirect, including disruptions in cellular redox balance and, as demonstrated here, the physical uncoupling of mitochondria from the ER (Chen et al, 2021). If this defect is a shared feature across ALS subtypes, it represents a crucial avenue for future research. While ER-mitochondria dissociation has been implicated in ALS pathogenesis, our findings establish a direct mechanistic link between the VAPB P56S mutation and disease phenotype, offering a clearer foundation for therapeutic development. Furthermore, the VAPB-PTPIP51 interaction may have broader implications in neurodegenerative diseases, as Liu et al recently demonstrated that α-synuclein can

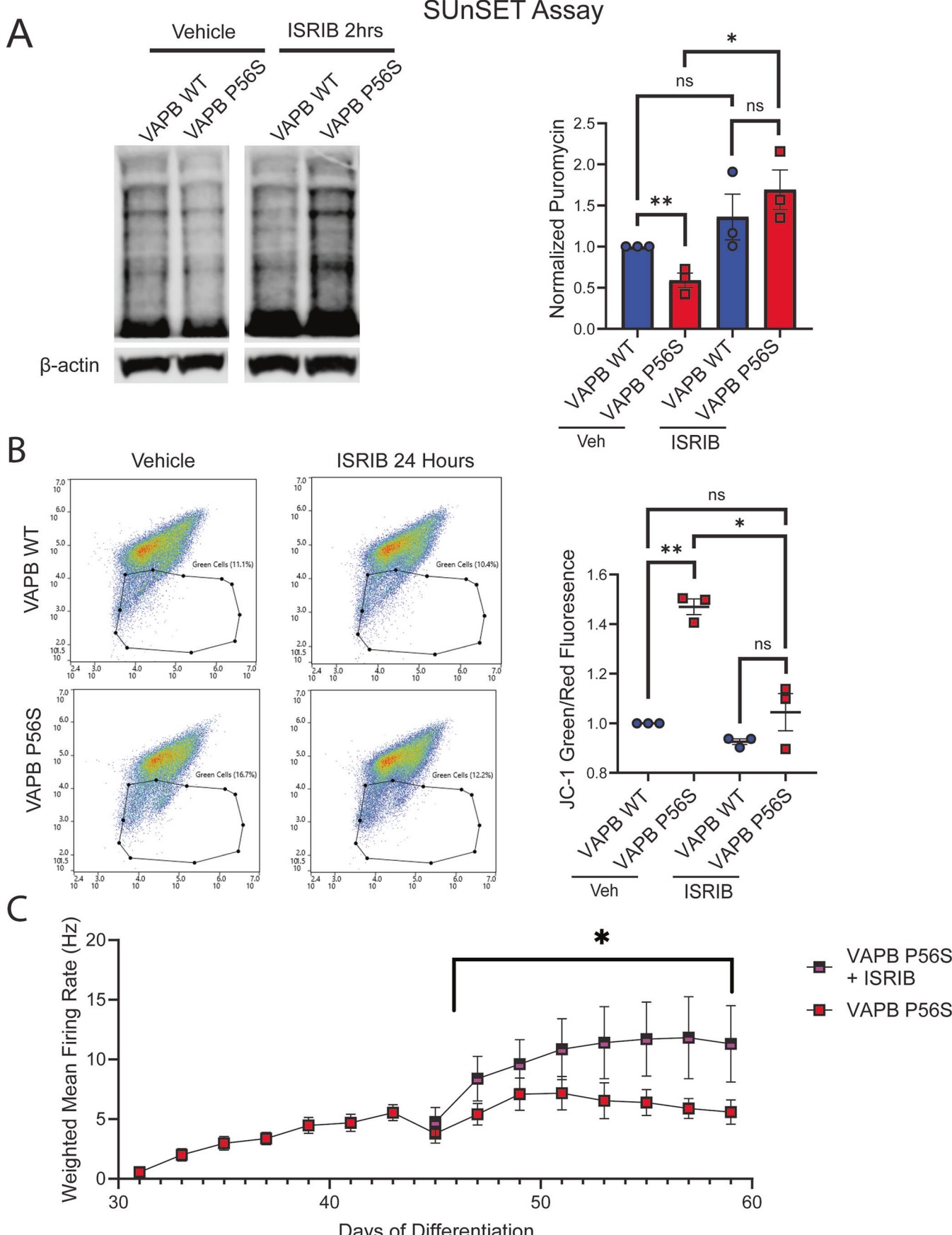

**Figure 5. ISR inhibition restores mRNA translation, mitochondrial membrane potential, and neuronal firing in doxycycline-inducible VAPB P56S iPSC-derived motor neurons.**

(A) SUnSET assay on day 35 of differentiation with vehicle control and after 2h ISRIB treatment. Data presented as mean ± SEM, $N = 3$ biological replicates, unpaired $t$-test: WT vs. P56S, $p = 0.0024$**; P56S vs. P56S + ISRIB, $p = 0.0125$*. (B) Scatter plot of day 60 motor neurons stained with JC-1 mitochondrial dye following 24 h ISRIB treatment. Data presented as mean ± SEM, $N = 3$ biological replicates, paired $t$-test: WT vs. P56S, $p = 0.0045$**; P56S vs. P56S + ISRIB, $p = 0.0482$*; two-way ANOVA for genotype and ISRIB treatment, $p = 0.0228$*. (C) Weighted mean firing rate through day 59 of differentiation. Data presented as mean ± SEM, $N = 4$ technical replicates (wells per condition). Two-way repeated measures ANOVA from day 45 onward, interaction effect $p = 0.0114$*. Source data are available online for this figure.

modulate VAPB-PTPIP51 binding, potentially linking this pathway to Parkinson's disease (Liu et al, 2025).

In conclusion, our findings suggest that increased cellular sensitivity and impaired recovery from ISR constitute the primary pathogenic pathway driving disease progression in ALS Type VIII. While further research is necessary to establish ISR involvement in other ALS subtypes and to advance therapeutic development, our results represent a significant step toward elucidating the molecular mechanisms underlying ALS and guiding the pursuit of effective therapeutic strategies.

# Methods

### Reagents and tools table

| Reagent/resource | Reference or source | Identifier or catalog number |
|---|---|---|
| **Experimental models** | | |
| VAPB P56S patient | Mitne-Neto et al, 2011 | 23705 |
| VAPB KO patient (23705) | This study | VAPB KO |
| Doxycycline-inducible VAPB WT (23705) | This study | VAPB WT |
| Doxycycline-inducible VAPB P56S (23705) | This study | VAPB P56S |
| VAPB P56S patient (Fibroblast) | Guber et al, 2018 | Pt VAPB P56S fibroblast |
| VAPB P56S patient (iPSC) | This study | Pt VAPB P56S |
| Isogenic VAPB WT | Synthego | Pt VAPB WT |
| Lenti-X HEK-293 | Clontech | Cat #NC9834960 |
| **Recombinant DNA** | | |
| pSpCas9(BB)-2A-GFP (PX458) | Addgene | Plasmid #48138 |
| pDONR™ 221 | Thermo Fisher | Cat #12536017 |
| pInducer20 | Addgene | Plasmid #44012 |
| pMDLg/pRRE | Addgene | Plasmid #12251 |
| pRSV-REV | Addgene | Plasmid #12253 |
| pCMV-VSV-G | Addgene | Plasmid #8454 |
| **Antibodies** | | |
| Mouse anti-Nanog (IF 1:250) | SantaCruz Biotechnology | Cat #sc-293121 |
| Rat anti-SOX2 (IF 1:500) | Thermo Fisher | Cat #14-9811-82 |
| Rabbit anti-OCT4 (IF 1:500) | Thermo Fisher | Cat #PA5-27438 |
| Mouse anti-Nestin (IF 1:200) | Thermo Fisher | Cat #MA1-110 |
| Chicken anti-GFAP (IF 1:500) | Abcam | Cat #ab4674 |

| Reagent/resource | Reference or source | Identifier or catalog number |
|---|---|---|
| Rabbit anti-Olig2 (IF 1:500) | Proteintech | Cat #13999-1-AP |
| Mouse anti-Islet 1/2 (IF 1:500) | DSHB | Cat #39.4D5 |
| Goat anti-chicken IgY (H + L) secondary antibody, Alexa Fluor™ 555 (IF 1:1000) | Thermo Fisher | Cat #A-21437 |
| Goat anti-rabbit IgG (H + L) cross-adsorbed secondary antibody, Alexa Fluor™ 647 (IF 1:1000) | Thermo Fisher | Cat #A-21244 |
| Donkey anti-mouse IgG (H + L) highly cross-adsorbed secondary antibody, Alexa Fluor™ 555 (IF 1:1000) | Thermo Fisher | Cat #A-31570 |
| Donkey anti-rabbit IgG (H + L) highly cross-adsorbed secondary antibody, Alexa Fluor™ 555 (IF 1:1000) | Thermo Fisher | Cat #A31572 |
| Donkey anti-rabbit IgG (H + L) highly cross-adsorbed secondary antibody, Alexa Fluor™ 488 (IF 1:1000) | Thermo Fisher | Cat #A21206 |
| Goat anti-mouse IgG (H + L) highly cross-adsorbed secondary antibody, Alexa Fluor™ 647 (IF 1:1000) | Thermo Fisher | Cat #A21236 |
| Goat anti-rat IgG (H + L) cross-adsorbed secondary antibody, Alexa Fluor™ 555 (IF 1:1000) | Thermo Fisher | Cat #A-21434 |
| Goat anti-mouse IgG (H + L) highly cross-adsorbed secondary antibody, Alexa Fluor™ 488 (IF 1:1000) | Thermo Fisher | Cat #A-11029 |
| Rabbit anti-VAPB (WB 1:1000) | Millipore Sigma | Cat #SAB1411626 |
| Rabbit anti-VAPB (WB 1:1000) | Proteintech | Cat #14477-1-AP |
| Rabbit anti-PTPIP51 (WB 1:1000) | Proteintech | Cat #20641-1-AP |
| Mouse anti-beta-actin (WB 1:5000) | Millipore Sigma | Cat #A2228 |
| Rabbit anti-ATF-4 (WB 1:500) | Cell Signaling Technologies | Cat #11815 |
| Rabbit anti-eIF2alpha (WB 1:1000) | Cell Signaling Technologies | Cat #9722 |
| Rabbit anti-phospho-eIF2alpha (EIF2S1 phospho S51) (WB 1:1000) | Abcam | Cat #ab32157 |
| Mouse anti-puromycin (WB 1:500) | Millipore Sigma | Cat #MABE343 |
| Rabbit anti-DELE1 (WB 1:500) | Proteintech | Cat #21904-1-AP |
| IRDye 800CW Donkey anti-Mouse (IHC: 1:10,000) | LI-COR | Cat #926-32212 |
| IRDye 680RD Donkey anti-Rabbit (IHC: 1:10,000) | LI-COR | Cat #926-68073 |
| **Oligonucleotides and other sequence-based reagents** | | |

| Reagent/resource | Reference or source | Identifier or catalog number |
|---|---|---|
| VAPB gRNA (w/*BbsI* restriction site at 5' end) | Eurofins Genomics | TACTGTGTGAGGTCCAACAG |
| VAPB GW forward primer | Eurofins Genomics | GGGGACAACTTTGTACAAA AAAGTTGGCATGGCGA AGGTGGA GCAGGTC |
| VAPB GW reverse primer | Eurofins Genomics | GGGGACAACTTTGTACAA GAAAGTTGGGCAAGGCAATCT TCC CAATAATTACACCAACG |
| VAPB forward (Sequencing) | Eurofins Genomics | GGTCCCTTCACCGATGTTGT |
| VAPB reverse (Sequencing) | Eurofins Genomics | CTACTGTCCAGGGGCCTTCT |
| pInducer/VAPB Exon 6 (qPCR forward) | Eurofins Genomics | TTTCAGCATTAGCCCCAACT |
| VAPB HA (qPCR reverse) | Eurofins Genomics | TCTGGGACGTCGTATGGGTA |
| Neomycin forward (qPCR) | Eurofins Genomics | AGACAATCGGCTGCTCTGAT |
| Neomycin reverse (qPCR) | Eurofins Genomics | AGTGACAACGTCGAGCACAG |
| RPL13 forward (qPCR) | Eurofins Genomics | CCTGGAGGAGAAGA GGAAAGAGA |
| RPL13 forward (qPCR) | Eurofins Genomics | TTGAGGACCTCTGTGT ATTTGTCAA |
| **Chemicals, enzymes, and other reagents** | | |
| Vitronectin (VTN-N) recombinant human protein, truncated | Thermo Fisher | Cat #A14700 |
| StemFlex™ Medium | Thermo Fisher | Cat #A3349401 |
| ReLeSR™ | Stemcell Technologies | Cat #100-0484 |
| Q5 High-Fidelity Polymerase | New England Biolabs | Cat #M0491 |
| Gateway™ BP Clonase™ II Enzyme mix | Thermo Fisher | Cat #11789020 |
| Gateway™ LR Clonase™ II Enzyme mix | Thermo Fisher | Cat #11791020 |
| DMEM (Dulbecco's Modified Eagle's Medium) | Corning | Cat #10-013-CV |
| Fetal Bovine Serum - Premium | BioTechne | Cat #S11150 |
| Kyfora Bio PEI 25 K™-Transfection Grade (Polyethylenimine, Linear, MW 25000) | Fisher Scientific | Cat #50-255-9825 |
| TRIzol™ | Thermo Fisher | Cat #15596026 |
| SuperScript™ IV VILO™ Master Mix | Thermo Fisher | Cat #11756050 |
| PowerTrack™ SYBR Green Master Mix for qPCR | Thermo Fisher | Cat #A46012 |
| DMEM/F12 | Thermo Fisher | Cat #11320033 |
| N2 Supplement (100X) | Thermo Fisher | Cat #7502048 |
| B-27™ Plus Supplement (50X) | Thermo Fisher | Cat #A3582801 |
| Penicillin-Streptomycin (10,000 U/mL) (PenStrep) | Thermo Fisher | Cat #15140122 |
| L-Ascorbic acid | Millipore Sigma | Cat #A4544-25G |
| Doxycycline hyclate | Millipore Sigma | Cat #D9891 |
| Dorsomorphin dihydrochloride | TOCRIS | Cat #3093 |

| Reagent/resource | Reference or source | Identifier or catalog number |
|---|---|---|
| TGF-beta RI Kinase Inhibitor VI (SB 431542) | Millipore Sigma | Cat #616461 |
| CHIR 99021 | Biogems | Cat #2520691 |
| Y-27632 Dihydrochloride | Biogems | Cat #1293823 |
| Retinoic acid | Millipore Sigma | Cat #R2625 |
| Smoothened Agonist (SAG) | Millipore Sigma | Cat #5666610 |
| Accutase | Innovative Cell Technologies | Cat #AT104-500 |
| Poly-L-ornithine hydrobromide | Millipore Sigma | Cat #P3655 |
| Laminin mouse protein, natural | Thermo Fisher | Cat #23017015 |
| Human brain-derived neurotrophic factor | GeminiBio | SKU# 300-104P-100 |
| Human ciliary neurotrophic factor | GeminiBio | SKU# 300-107P-100 |
| Human glial-derived neurotrophic factor | GeminiBio | SKU# 300-121P-100 |
| DAPT | TOCRIS | Cat #2634 |
| Accumax | Innovative Cell Technologies | Cat #AM105 |
| Paraformaldehyde | Fisher Scientific | Cat #AC416780250 |
| Triton™ X-100 | Millipore Sigma | Cat #X100 |
| Bovine serum albumin | Millipore Sigma | Cat #A9418-50G |
| Hoechst 33258 | Thermo Fisher | Cat #H1398 |
| Puromycin dihydrochloride | Thermo Fisher | Cat #A11138-03 |
| Vector Laboratories VECTASHIELD Antifade Mounting Medium | Cole-Parmer | Item #EW-93952-23 |
| RIPA Lysis and Extraction Buffer | Thermo Fisher | Cat #89901 |
| Pierce Protease Inhibitor Tablets | Thermo Fisher | Cat #A32955 |
| Pierce BCA assay kit | Thermo Fisher | Cat #23227 |
| Bolt 4–12% bis-tris plus gels | Thermo Fisher | Cat #NW04120BOX |
| Immobilon-P PVDF membrane | Millipore Sigma | Cat #IPVH00010 |
| Intercept (PBS) blocking buffer | LI-COR | Cat #927-60001 |
| Polysorbate 20 | Fisher Scientific | Cat #BP337-100 |
| MitoProbe JC-1 Assay Kit | Thermo Fisher | Cat #M34152 |
| Dynabeads™ Protein A for Immunoprecipitation | Thermo Fisher | Cat #10001D |
| Pierce Protease Inhibitor Tablets | Thermo Fisher | Cat #A32955 |
| Tunicamycin | TOCRIS | Cat #3516 |
| ISRIB | Millipore | Cat #SML0843 |
| **Software** | | |
| CFX Maestro | Bio-Rad | N/A |

| Reagent/resource | Reference or source | Identifier or catalog number |
|---|---|---|
| FIJI v2.14.0 | https://imagej.net/software/fiji/ | N/A |
| Image Studio Lite | LI-COR | N/A |
| Empiria Studio v3.2.0.186 | LI-COR | N/A |
| WOLFViewer | NanoCellect | N/A |
| AxIS Neural Module | Axion Biosystems | N/A |
| Neural Metrics | Axion Biosystems | N/A |
| Prism v10.4.1 | GraphPad | N/A |
| Other | | |
| Epi5™ Episomal iPSC Reprogramming Kit | Thermo Fisher | Cat #A15960 |
| GenClone 25-260, TC Treated Dishes, 60 × 15 mm | Genesee Scientific | Cat #25-260 |
| Bio-Rad CFX96 Thermocycler | Bio-Rad | N/A |
| GenClone 25-202, TC Treated Dishes, 100 × 20 mm Vented, 60.8 cm², Gripping Ring | Genesee Scientific | Cat #25-202 |
| GenClone 25-105, six-well cell culture plates flat flat-bottom wells, TC treated | Genesee Scientific | Cat #25-105 |
| Nunc™ Lab-Tek™ II CC2™ Chamber Slide System | Thermo Fisher | Cat #154739 |
| Biorupter Pico sonication device | Diagenode | Cat #B01080010 |
| TransBlot Turbo transfer station | Bio-Rad | Cat #1704150 |
| Odyssey FC Imager | LI-COR | N/A |
| WOLF G1 Sorter | NanoCellect | N/A |
| CytoView MEA 48-well plate | Axion Biosystems | Cat #M768-tMEA-48B |
| Maestro Pro | Axion Biosystems | N/A |
| AxioObserver | Zeiss | N/A |
| NEON™ Transfection System | Thermo Fisher | N/A (Old Model, Discontinued) |

## Culturing human induced pluripotent stem cells

The VAPB WT and P56S doxycycline-inducible lines were generated from patient fibroblasts with the P56S mutation through the use of the Epi5™ Episomal iPSC Reprogramming Kit (Thermo Fisher) (Mitne-Neto et al, 2011). The iPSC lines used in Fig. 6 were generated from a second culture of VAPB P56S fibroblasts obtained from the Grunseich lab at the National Institute of Health and were similarly reprogrammed here at CWRU (Guber et al, 2018). This patient VAPB P56S line was sent to Synthego for CRISPR-Cas9 modification to create the isogenic patient VAPB WT line. Cultures were tested bi-weekly for mycoplasma and maintained in an antibiotic-free medium. The iPSCs were maintained on 6 cm cell culture plates (Genesee) coated with vitronectin (VTN-N) (Thermo Fisher; 5 ug/mL) in StemFlex medium (Thermo Fisher). To pass the iPSC colonies for maintenance and expansion, 1 mL ReLeSR dissociation reagent (Stemcell Technologies) was added to the culture plate for 30 s at room temperature, then the ReLeSR was aspirated until a thin film remained on top of the cells. Cells were

then incubated for 3–5 min at 37 °C. Cells were then gently lifted off and resuspended in StemFlex medium and distributed to new 6-cm plates.

## VAPB knockout

To target the VAPB Exon 2, a guideRNA (VAPB gRNA) was designed and inserted into pSpCas9(BB)-2A-GFP (PX458) (Addgene #48138) using the protocol described in the literature by the designer of the plasmid (Ran et al, 2013). Patient-derived iPSC were transfected using a NEON Transfection system, and sorted using a NanoCellect WOLF to isolate green-fluorescing cells. Colonies were then screened using DNA sequencing the find cells with insertions and/or deletions (Fig. EV1A), and the knockout was confirmed through Western Blot (Fig. EV2A).

## tetON VAPB vector synthesis and iPSC transduction

The vector containing VAPB WT and P56S genes under the control of a tetracycline response element were created through the use of Invitrogen's Gateway Cloning System, and the VAPB KO iPSC were transduced. attB flanked VAPB was created through PCR amplification of the VAPB gene using the VAPB GW forward and reverse primers using a Q5 High-Fidelity Polymerase (New England Biolabs). VAPB was then inserted into Gateway™ pDONR™221 Vector (Thermo Fisher) using the Gateway™ BP Clonase™ II Enzyme mix (Thermo Fisher) following the protocol as described by the manufacturer. VAPB was then placed into the pInducer20 (addgene #44012) using Gateway™ LR Clonase™ II Enzyme mix (Thermo Fisher) following the protocol as described by the manufacturer. The pInducer20+VAPB vector was then packaged into a lentiviral vector, and the VAPB KO iPSC were transduced as previously described in the literature (Chen et al, 2024). Briefly, we created lentiviruses using PEI transfection of Lenti-X HEK-293 cells (Clontech). HEK-293 cells were cultured on 15 cm plates with DMEM (Corning) with 4.5 g/L of glucose + 10% FBS (BioTechne) (HEK media) until they reached about 50% confluence. For every 12 plates of HEK-293 cells, 500 uL of blank DMEM was mixed with pInducer20 + VAPB (12.2 ugs), pMDLg/pRRE (Addgene #12251) (8.1 ugs), pRSV-REV (Addgene #12253) (3.1 ugs), pCMV-VSV-G (Addgene #8454) (4.1 ugs), and PEI (Fisher Scientific; 1 ug/uL). The solution was incubated with HEK-293 cells overnight at 37 °C, then the medium was replaced with fresh HEK media and incubated for 3 days. HEK medium was then collected after 3 days and centrifuged at 500×g for 10 min. The supernatant was transferred to a fresh conical tube, and the Lenti-X concentrator was added per manufacturer's instructions (Clontech). The supernatant was centrifuged at 1500×g for 45 min or until the viral pellet became visible. The supernatant was then removed, and the viral pellet was resuspended in 500 uL of DMEM/F12. 100 uL of the viral suspension is added per 1–2 million cells. VAPB doxycycline-dependent expression was then assayed through qPCR and Western Blot (Figs. EV1D and EV2A).

## Quantitative PCR

RNA was extracted using TRIzol Reagent (Thermo Fisher) and the procedure was performed according to their provided protocol. cDNA was generated using SuperScript™ IV VILO™ Master Mix

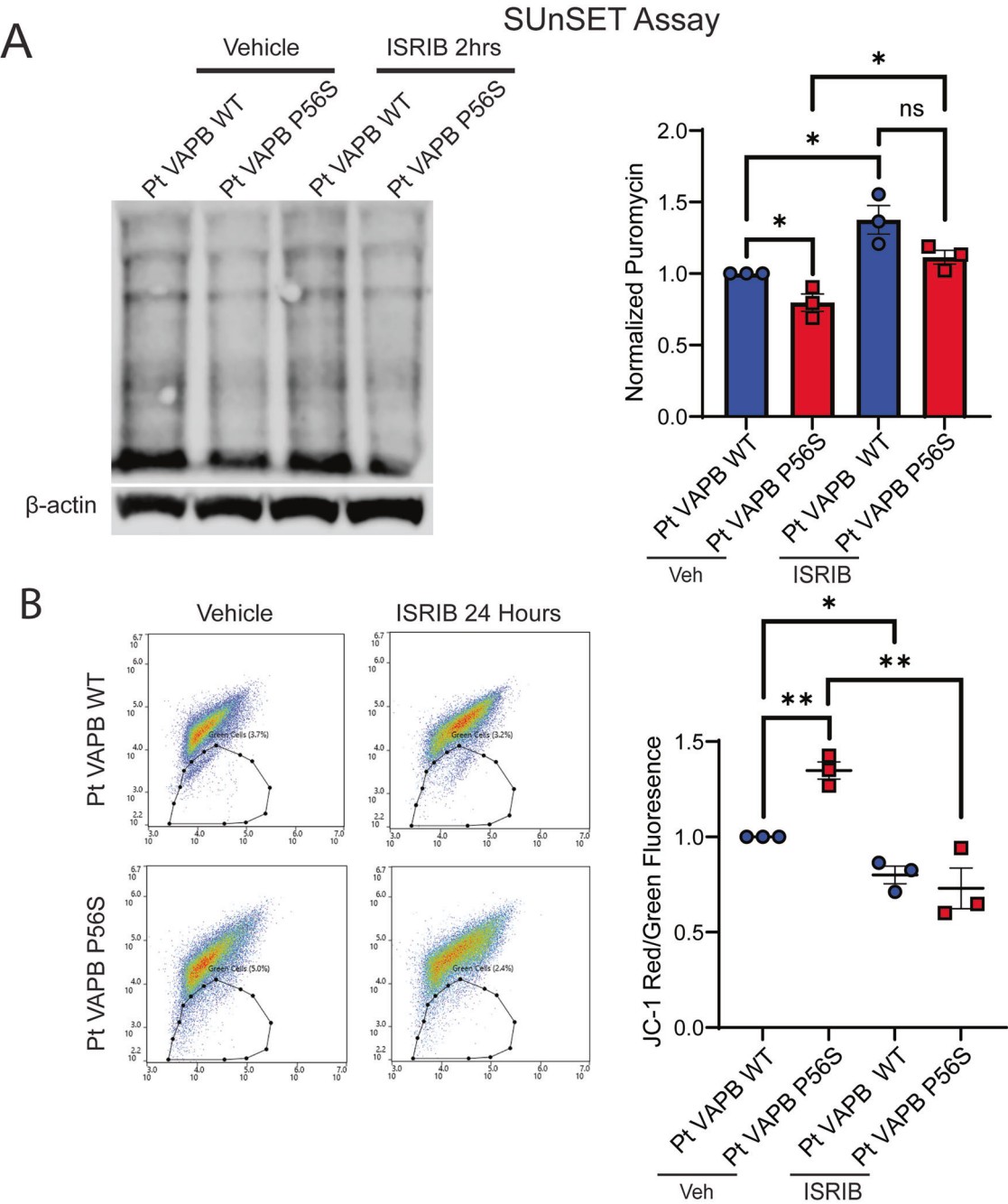

**Figure 6.  ISR inhibition restores mRNA translation, mitochondrial membrane potential, and neuronal firing in ALS patient (Pt) iPSC-derived motor neurons.**

(A) SUnSET assay on day 35 of differentiation with vehicle control and after 2 h ISRIB treatment. Data presented as mean ± SEM, $N = 3$ biological replicates, unpaired t-test: Pt WT vs. Pt P56S, $p = 0.0296^*$; Pt P56S vs. Pt P56S + ISRIB, $p = 0.0155^*$; Pt WT vs. Pt WT + ISRIB, $p = 0.0201^*$. (B) Scatter plot of day 35 motor neurons stained with JC-1 mitochondrial dye following 24 h ISRIB treatment. Data presented as mean ± SEM, $N = 3$ biological replicates, paired t-test: Pt WT vs. Pt P56S, $p = 0.0015^{**}$; Pt P56S vs. Pt P56S + ISRIB, $p = 0.006^{**}$; Pt WT vs. Pt WT + ISRIB, $p = 0.0122^*$. Two-way ANOVA for genotype and ISRIB treatment, $p = 0.0034^{**}$. Source data are available online for this figure.

(Thermo Fisher) and the procedure was performed according to their provided protocol. qPCR was performed using PowerTrack™ SYBR Green Master Mix for qPCR (Thermo Fisher), and the procedure was performed according to the provided protocol, using a Bio-Rad CFX96 thermocycler, samples were run in triplicate.

Quantification was performed using CFX Maestro software from Bio-Rad. VAPB was normalized to Neomycin and RPL3 using the software, and the resultant Expression values were then graphed along with the provided SEM, per standards in the field (Livak and Schmittgen, 2001; Wong and Medrano, 2005).

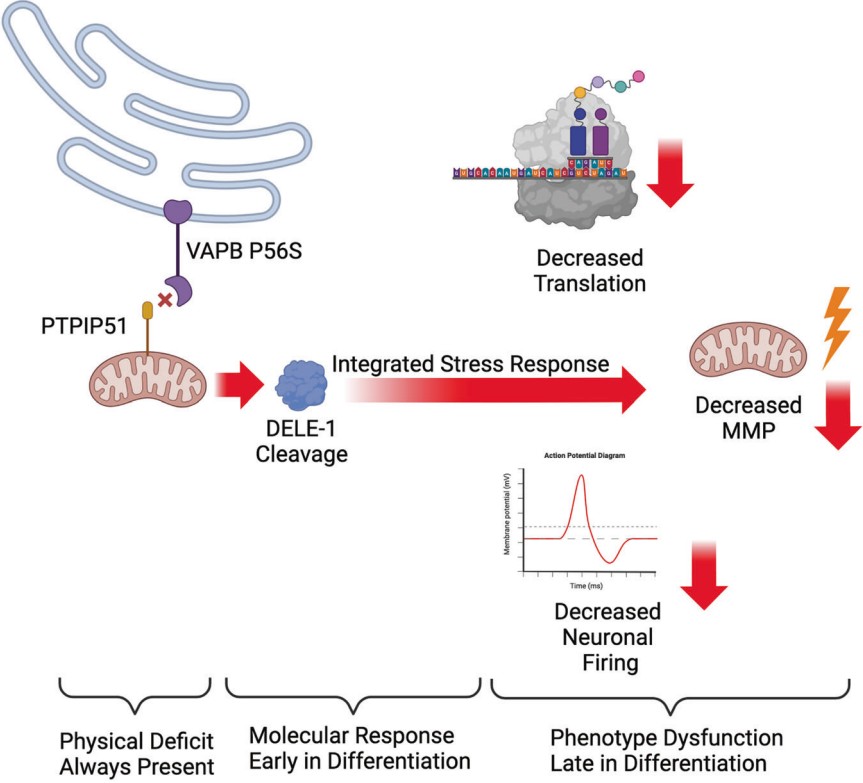

**Figure 7. Proposed model of ALS pathogenesis.**

Schematic representation of the proposed pathogenic mechanism. Created in BioRender. https://BioRender.com/rdfmnqy.

## Generating motor neurons from human iPSCs

Motor neurons were cultured as described in Chen et al, 2024 with minor modifications (Chen et al, 2024). Briefly, iPSCs were cultured on 6 cm treated plates until they reached about 75% confluence. The base medium used for motor neuron differentiation was DMEM/F12 with 0.5x N2 (Thermo Fisher), 0.5x B-27 (Thermo Fisher), PenStrep (Thermo Fisher; 100 ug/mL), and ascorbic acid (Millipore; 0.2 mM). This is referred to as motor neuron media. For the doxycycline-inducible lines, doxycycline (Millipore) was added to the motor neuron starting on day 1 at a concentration of 0.2 ug/mL. On days 1–6 of motor neuron differentiation, the culture medium was changed to motor neuron media with dorsomorphin (TOCRIS; 1 uM), SB 431542 (Millipore; 10 uM), CHIR 99021 (Biogems; 3 uM), and Y-27632 (Biogems; 5 uM). On day 6, cells were split 1:3 from 6-cm plates to 10-cm plates (Gennesse). From day 7 to day 15, cells were maintained in the motor neuron media with dorsomorphin (1 uM), SB 431542 (10 uM), retinoic acid (Millipore; 1.5 uM), SAG (Millipore; 200 nM), and Y-27632 (10 uM). On day 15 of differentiation, cells were dissociated with Accutase (Innovative Cell Technologies) and replated in six-well plates (Genesee) coated with poly-ornithine (Millipore; 10 ug/mL) and Laminin (Thermo Fisher; 2.5 ug/mL). $1 \times 10^6$ cells were plated per well and cultured in motor neuron media with human brain-derived neurotrophic factor (BDNF; GeminiBio; 2 ng/mL), human ciliary neurotrophic factor (CNTF; GeminiBio; 2 ng/mL), human glial-derived neurotrophic factor

(GDNF; GeminiBio; 2 ng/mL), retinoic acid (1.5 uM), SAG (200 nM), and Y-27632 (5 uM). Cells were left in this medium until day 21. On day 22, the media was changed to the motor neuron media with BDNF (2 ng/mL), CNTF (2 ng/mL), GDNF (2 ng/mL), DAPT (TOCRIS; 2 uM), and Y-27632 (5 uM). DAPT was removed on day 25 of differentiation. On day 25, motor neurons can be dissociated and replated or maintained with media changes every 1–2 days.

Motor neurons were dissociated using a 1:1 dissociation solution of Accutase and Accumax (Innovative Cell Technologies Inc.) were counted using Trypan Blue to ensure a survival rate above 80%. Cells were incubated in the dissociation solution for 10–15 min at 37 °C. After the time had elapsed, the plates were tapped against a surface to lift the cells. Cells were then triturated a maximum of five times and passed through a 70-uM strainer. It is important to minimize the number of triturations to maximize cell viability. Cells were then centrifuged at 200×g for 2 min. Motor neurons and motor neuron progenitor cells can also be plated onto glass surfaces coated with poly-ornithine (100 ug/mL) and Laminin (5 ug/mL). Motor neurons could also be frozen in homemade freezing media. For 50 mL of motor neuron freezing media: 25 mL of motor neuron media, 20 mL of FBS, 5 mL of DMSO, and 10 uM of Y-27632.

## Immunofluorescence

Cells were plated onto an 8-chamber Nunc™ Lab-Tek™ II CC2™ Chamber Slide System (Thermo Fisher), coated with VTN-N (5 ug/

mL) for iPSC or Laminin (5 ug/mL) for iPSC-derived motor neuron progenitors (day 15). Once cells reached the desired day of differentiation (or confluency in the case of iPSC), the media was aspirated, and cells were gently rinsed twice with PBS. 4% paraformaldehyde (PFA; Fisher Scientific) was then added to the wells and incubated at room temperature for 15 min. PFA was then aspirated, and the cells were gently washed with PBS three times. 0.25% Triton X-100 (Millipore) was then added to the cells and incubated at room temperature for 15 min. The 0.25% Triton X-100 was then aspirated, and the cells were washed gently with PBS three times. 3% bovine serum albumin (BSA; Millipore) in PBS was then added to the cells and incubated for 3 h at room temperature. The 3% BSA was then aspirated, and the primary antibody diluted in 3% BSA was added to the cells and incubated in the dark, at 4 °C overnight. The primary antibody was then removed, and the cells were washed gently with PBS three times. Secondary antibody was then diluted in 3% BSA, added to the cells, and incubated in the dark, at room temperature for 1 h. Secondary antibody solution was aspirated, and Hoechst 33258 (Thermo Fisher; 2 ug/mL) was then added and incubated for 10 min at room temperature. Hoechst 33258 was removed, and the cells were gently washed with PBS three times. The walls of the chamber slide were then removed with the provided tool, and any remaining liquid was carefully aspirated off the surface of the slide. VECTASHIELD anti-fade mounting medium (Cole-Parmer) was then added dropwise to the slide, and a coverslip laid over. Nail polish was then used to seal the coverslip to the slide, and the slide was then imaged using the AxioObserver with Apotome (Zeiss) and quantified using FIJI.

## Surface sensing of translation assay

SUnSET assay was performed based on a previously published protocol with some slight modifications (Schmidt et al, 2009). Briefly, we added puromycin dihydrochloride (Thermo Fisher) at 10 ug/mL to the motor neuron media and incubated at 37 °C for 10 min. After incubation, we removed the puromycin, washed the cells once with motor neuron media, and incubated them in day 25+ motor neuron media for 45 min at 37 °C. After this incubation, cells were harvested and proteins extracted, with Western Blots being performed as described below, with the caveat that total protein was not quantified and equalized across samples; instead, equal volumes of protein extract were loaded, and the samples were normalized through beta-actin staining on a separate gel.

## Western blot

For all the western blots in this study, cells were harvested in Pierce RIPA Lysis and Extraction Buffer (Thermo Fisher) (150 uL per well of a six-well plate) with Pierce Protease Inhibitor Tablets (Thermo Fisher) added. Cells were then kept at −80 °C until ready for protein extraction. Cell pellets were thawed on ice and then sonicated with a Bioruptor Pico sonication device (Diagenode) for 10 min in a 30 s on-off cycle at 4 °C. Solutions were homogenized through repeated pipetting if needed, and then centrifuged at 20,000 rcf for 25 min at 4 °C. Supernatant was collected and used to perform a Pierce BCA assay (Thermo Fisher) to quantify total protein. Western blots were run using 20 ug of total protein per well on Bolt 4–12% bis-tris plus gels (Thermo Fisher). Gels were

then transferred to Immobilon-P PVDF membrane (Millipore) using a Bio-Rad TransBlot Turbo transfer station at 2.5 A, 25 V for 10 min. The membrane was then blocked for 1 h in a 4% milk, 1% polysorbate 20 (Tween-20; Fisher Scientific) in PBS solution on a shaker. The milk buffer was then removed, and primary antibodies suspended in Intercept (PBS) blocking buffer (LI-COR) were added. Membranes were incubated in primary antibody solution on a shaker overnight at 4 °C. After incubation, the primary antibody solution was removed, and the membrane was rinsed with 1% polysorbate 20 (Tween-20; Fisher Scientific) in PBS (PBS-T) three times, then placed on a shaker in PBS-T for 5 min. This wash was repeated three times, and then secondary antibodies (LI-COR IRDye 800CW Donkey anti-Mouse and IRDye 680RD Donkey anti-Rabbit) diluted in 4% milk in PBS-T (1:10,000) were added to the membrane and incubated for 1 h at room temperature on a shaker. The membrane was then washed as previously in PBS-T and imaged on a LI-COR Odyssey FC imager. Quantification was performed using the Image Studio Lite (LI-COR) or Empiria Studio Software (LI-COR). Unless otherwise noted, values were normalized to β-actin within each sample, with a designated control (typically wild-type) set to 1; all other values were expressed as fold change relative to this control.

## JC-1 assay

The JC-1 assay was performed according to the protocol outlined in the MitoProbe JC-1 Assay Kit (Thermo Fisher) with a few minor modifications. Briefly, motor neurons were dissociated using a 1:1 Accutase:Accumax solution (1 mL per well in a six-well plate), incubated at 37 °C for 15 min. Once the neurons had lifted from the plate, 1 mL of day 25+ motor neuron media was added, and the neurons were gently pipetted up and down one to two times to homogenize them. The cells were then put through a 70-uM mesh strainer, counted, and centrifuged at 200 rcf for 2 min. Supernatant was removed, and the cells were resuspended at a concentration of $1 \times 10^6$ cells/mL in day 25+ motor neuron media. About 1 mL of the cells was then placed into a 1.5 mL tube, and 8 uL of 200-uM JC-1 dye was added to each tube (increased to 10 uL for day 60 motor neurons). A separate aliquot of cell suspension was also incubated with 4 uL of the supplied 50 mM CCCP for 15 min prior to JC-1 dye addition, to act as a positive control and ensure the JC-1 dye was correctly detecting low MMP populations. These were then incubated at 37 °C, 45 min for day 35 neurons, and 75 min for day 60 neurons, as some day 60 motor neurons remained unstained after only 45-min incubation. After incubation, the cells were centrifuged at 200 rcf for 90 s and the supernatant removed. Cells were then washed with 1 mL of sterile PBS. This wash step was repeated twice more with the cells being resuspended in 750 uL of sterile PBS filtered through a 0.22-uM filter on the final wash. The resulting solution was then run through a WOLF G1 (Nanocellect) sorter with the gains for FL1, FL2, and FL3 all set to 200 mV. Data were compensated with FL1 to FL2 spillover set to 22% and FL2 to FL1 spillover set to 6%. Samples were run until at least 50,000 events were recorded, and green fluorescent populations were then gated out by eye, assisted by the population density heat mapping function to ascertain population centers. The percentage of the total population fluorescing green was then normalized to the WT control of each individual biological replicate.

## Proteomics

About 20 uL of Dynabeads Protein A (Thermo Fisher) were incubated with 4 ug of VAPB antibody overnight. Cells were then harvested with 300 uL of Martina lysis buffer (25 mM HEPES pH 7.4, 150 mM NaCl, 5 mM EDTA, 1% Triton X-100) with Pierce Protease Inhibitor Tablets (Thermo Fisher) added. The cells were then homogenized using a 5 mL syringe and 25G needle, then centrifuged at 20,000 rcf for 25 min at 4 °C. Supernatant was collected and used to perform a Pierce BCA assay (Thermo Fisher) to quantify total protein. A magnetic rack was then used to hold the beads (now conjugated to the antibody) in place while the antibody solution was removed, and 1 mg of total protein extract was added to the beads. The beads-protein solution was then incubated for 2 h at 4 °C on a rotator. Then, using a magnetic rack, the beads were held in place, and the supernatant was removed, and the beads were washed with lysis buffer three times and submitted to the Proteomics Core at the Lerner Research Institute. The method used for identification was adopted from previous literature (Mohammed et al, 2016). The mass spectrometry proteomics data have been deposited to the ProteomeXchange Consortium via the PRIDE partner repository with the dataset identifier PXD065524 and 10.6019/PXD065524 (Perez-Riverol et al, 2025). The identification of VAPB binding partners via mass spectrometry was performed with one biological sample, while the validation of VAPB-PTPIP51 binding via co-immunoprecipitation and Western blot was performed with three separate biological replicates.

## Multi-electrode array recordings

Motor neurons were dissociated on day 25 of differentiation and replated onto a CytoView 48-well MEA plate (Axion Biosystems) coated with poly-ornithine (100 ug/mL) and laminin (5 ug/mL), $1 \times 10^5$ cells/well. iPSC-derived MNs from a single well of a six-well plate thawed as day 15 MNP were dissociated and plated across eight wells of the MEA plate. Each point on the graph is an average of the weighted mean firing rate of those eight wells, normalized for cell count across genotypes, obtained after all firings were recorded by dissociating two wells per line, counting and averaging the cell numbers, and then normalizing all firings by the ratio of cell number between WT and P56S. Wells with no firing detected were excluded from quantification. Media was changed every other day, with recordings being taken with the Maestro Pro (Axion Biosystems) for at least 5 min, 1 h post-media change, beginning by day 30 of differentiation. Each recording was performed at 37 °C and 5% $CO_2$ using the AxIS Software Spontaneous Neural Configuration for spontaneous activity. Recordings were then processed with Axion Biosystems' Neural Metrics Tool, and values for each well were exported and analyzed. Bursts were identified using an interspike interval (ISI) threshold requiring a 5-spike minimum and 25-ms maximum ISI.

## Electron microscopy

Motor neurons were cultured as normal, and then roughly 7 days before fixation, they were dissociated and replated onto a 12 mm Snapwell™ Insert with 0.4 µm Pore Polyester Membranes (Corning). They were then fixed at their respective timepoints by removing the growth medium, washing the plate, and adding 2–3 mL of fixative

solution consisting of 2.5% glutaraldehyde in cacodylate buffer (pH 7.3), to the top and bottom of the membrane. The cells were incubated in the fixative solution for 1 h at room temperature. The fixative solution was then removed, and a fresh 2–3 mL was once again added to the top and bottom of the membrane and incubated for another hour at room temperature. The fixative solution was removed, and then the membrane was washed with PBS three times, leaving the PBS on the membrane for 5 min each time. The wells with membrane inserts were then completely filled with PBS and kept at 4 °C until they were transported to the cryo-electron microscopy core facility at the Cleveland Center for Membrane and Structural Biology for processing and imaging. The specimen was postfixed in ferrocyanide-reduced 1% osmium tetroxide (Karnovsky, 1971). After a soak in acidified uranyl acetate, the specimen was dehydrated in ethanol, passed through propylene oxide, and embedded in Embed-812 (Electron Microscopy Science) (Tandler, 1990). Sections were cut in a horizontal plane parallel to that of the membrane to provide panoramic views of the cells. Thin sections were stained first with acidified uranyl acetate in 50% methanol, then with the triple lead stain of Sato as modified by Hanaichi et al (Hanaichi et al, 1986; Tandler, 1990). These sections were examined in a FEI Tecnai Spirit (T12) with a Gatan US4000 4kx4k CCD. The cryo-electron microscopy core facility provided a series of images and magnifications and all the unique images at the highest magnification that were provided to us for each of the following categories: D35 WT: 13 unique images, D35 P56S: 21 unique images, D60 WT 13 unique images, D60 P56S: 18 unique images were included in the analysis. All images for a given line come from a single well of a 12 mm Snapwell™ Insert with 0.4 µm pore polyester membranes (Corning). No indication of cell grouping or sampling techniques was provided with the images; therefore, the images were quantified as a random sampling of the culture. Images were then blinded, and FIJI was used to quantify. Mitochondria were identified through the visual presence of cristae and a double membrane. The perimeter of these objects was quantified using the free draw tool, and the length of ER membranes within 50 nm were then quantified, with the final measurement graphed being the percentage of the perimeter of any given mitochondria in contact with the ER, an aggregate of any number of visually separate ER-MAMs, as it is impossible to be certain any given ER MAM is or is not contiguous outside the plane of the image (Cosson et al, 2012; Csordás et al, 2010; Stoica et al, 2014). Each data point on the graph is a single mitochondrion, data were gathered from multiple cells across multiple unique images provided by the Core from a single biological replicate given to them.

## Tunicamycin

Tunicamycin (TOCRIS) was resuspended in DMSO at a concentration of 5 mg/mL. When dosing cells, it was added to the medium at a final working concentration of 7.5 ug/mL, with this concentration being used for all results shown here.

## ISRIB

ISRIB (Millipore) was resuspended in DMSO at a concentration of 5 mg/mL. When dosing cells, it was added to the medium at a final working concentration of 5 ug/mL, apart from the neuronal firing

rescue experiment (Fig. 5C) for which it was added for a final concentration of 15 ug/mL. ISRIB was added every 8 h for the JC-1 ISRIB rescue experiment (Fig. 5B) and every 24 h for MEA recordings (Fig. 5C).

## Statistical analysis

All statistical analyses were performed using GraphPad Prism v10.3.1. All experiments were conducted with three independent motor neuron differentiations as biological replicates, except for the ER-mitochondria associated membrane (ER MAM) analysis, where a single population of iPSC-derived motor neurons per condition was submitted for electron microscopy. Multiple cells and mitochondria were analyzed per condition from the images provided by the core facility. Statistical tests are specified in the corresponding figure captions, and a *p* value of <0.05 was considered statistically significant. Sample sizes are consistent with established standards in the field for iPSC-derived neuronal studies.

## Data availability

The datasets produced and used in this study are available in the following databases.

Protein Interaction AP-MS Data: PRIDE PXD065524 (http://www.ebi.ac.uk/pride/archive/projects/PXD065524).

The source data of this paper are collected in the following database record: biostudies:S-SCDT-10_1038-S44321-025-00279-3.

## Peer review information

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

## Acknowledgements

We thank all lab members and the CWRU motor neuron group meetings for frequent discussions and valuable input. We would also like to thank Dr.

Belinda Willard and the Proteomics and Metabolomics Core at the Lerner Research Center for performing the mass spectrometry experiment to identify the VAPB interactome (Fig. 2), and Dr. Hisashi Fujioka and the Cryo-Electron Microscopy Core at Cleveland Center for Membrane and Structural Biology for imaging the ER-Mitochondria associated membranes (Figs. 3A,B and EV6A,B).

This work was supported by NINDS K01NS116119 to (HCM), R01NS123524 to (AES), and DK060596 to (MH).

## Author contributions

**Curran Landry**: Conceptualization; Data curation; Formal analysis; Validation; Investigation; Methodology; Writing—original draft; Writing—review and editing. **James P Costanzo**: Data curation; Formal analysis. **Miguel Mitne-Neto**: Conceptualization; Visualization; Writing—review and editing. **Mayana Zatz**: Conceptualization. **Ashleigh E Schaffer**: Conceptualization; Supervision; Visualization; Writing—review and editing. **Maria Hatzoglou**: Conceptualization; Supervision; Investigation; Writing—review and editing. **Alysson R Muotri**: Conceptualization. **Helen C Miranda**: Conceptualization; Resources; Data curation; Software; Formal analysis; Supervision; Funding acquisition; Investigation; Methodology; Writing—original draft; Project administration; Writing—review and editing.

Source data underlying the figure panels in this paper may have individual authorship assigned. Where available, figure panel/source data authorship is listed in the following database record: biostudies:S-SCDT-10_1038-S44321-025-00279-3.

## Disclosure and competing interests statement

HCM and ARM serve on the Scientific Advisory Board of Axion BioSystems, whose technology was used in this study. The authors declare no competing interests.

# Expanded View Figures

**Figure EV1. Characterization of doxycycline-inducible VAPB iPSCs.**

(A) Next-generation sequencing of patient iPSCs following CRISPR-Cas9-mediated mutagenesis revealed two frameshift mutations: a 2 bp deletion in the WT VAPB allele and a 1 bp insertion in the P56S allele. (B) Sanger sequencing of VAPB WT and VAPB P56S lines after transduction with doxycycline-inducible constructs; the P56S mutation is highlighted in red. (C) Immunofluorescence analysis confirming robust expression of pluripotency markers Nanog, SOX2, and Oct4 in all inducible lines. $N = 3$ biological replicates, minimum of three technical replicates. (D) Quantitative PCR of HA-tagged VAPB transcripts in doxycycline-inducible iPSC lines. Expression was normalized to Neomycin and RPL3. Data were shown as mean ± SEM; $N = 3$ qPCR biological replicates per line, per time point.

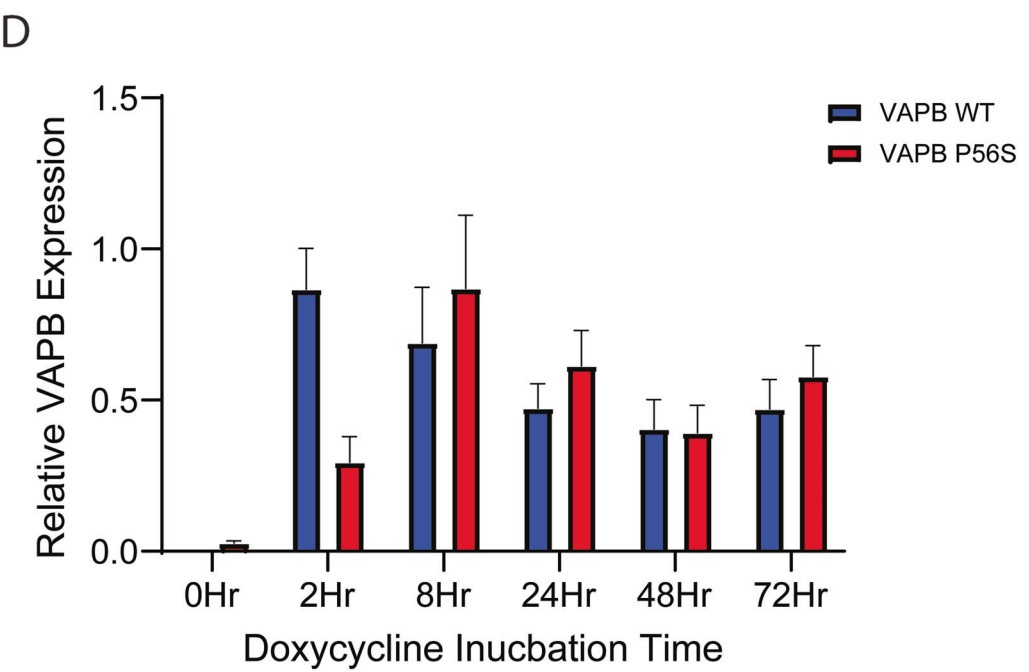

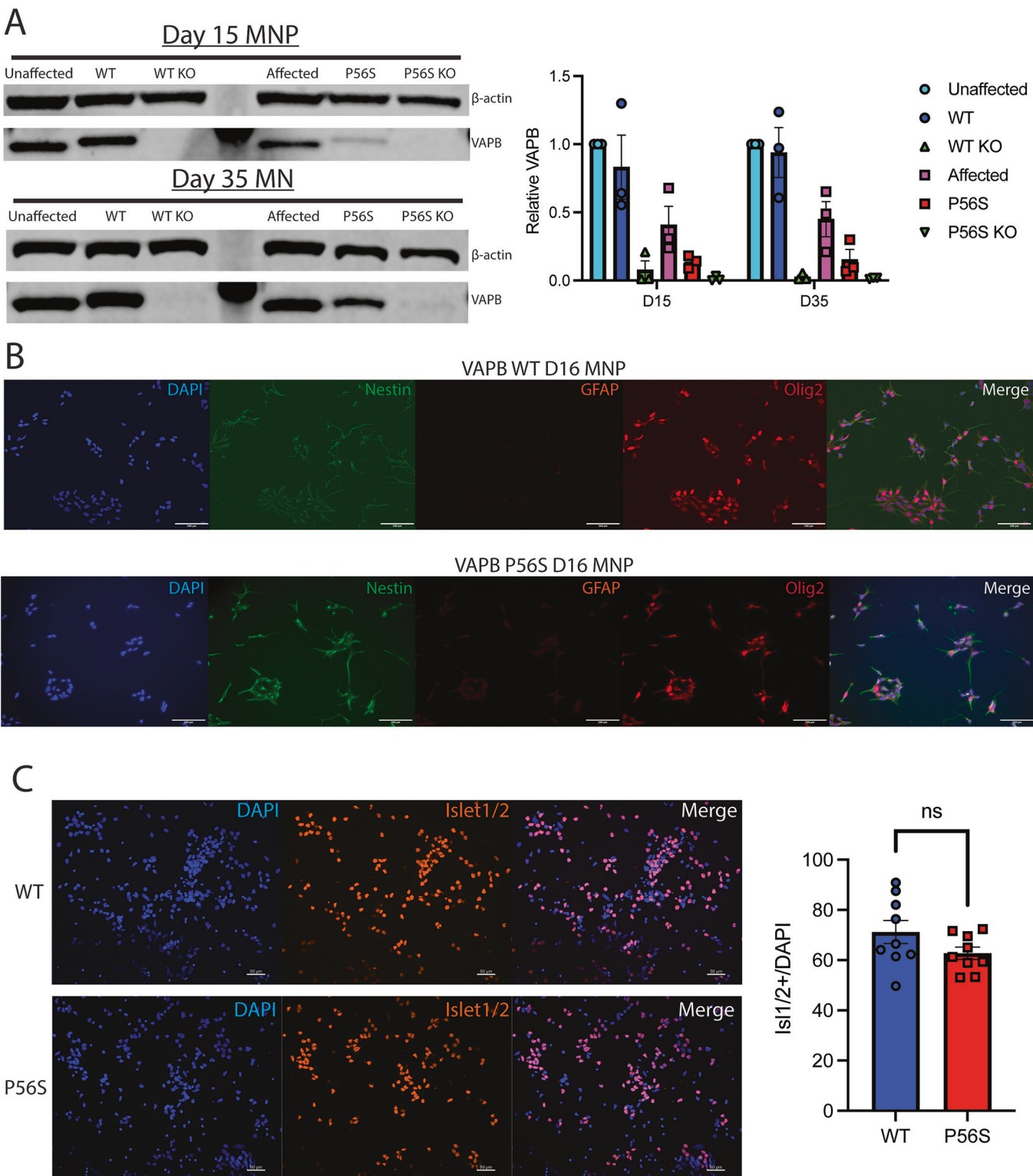

**Figure EV2. Characterization of doxycycline-inducible VAPB iPSC-derived motor neurons.**

(A) Western blot analysis of VAPB expression at days 15 (motor neuron progenitors, MNP) and 30 of differentiation. "Unaffected" and "Affected" refer to familial patient iPSC lines, with inducible lines generated from the Affected line. Data were presented as mean ± SEM; $N = 3$ biological replicates. (B) Immunofluorescence on day 15 showing expression of motor neuron progenitor markers Nestin and Olig2, with no detectable expression of the glial marker GFAP. $N = 3$ biological replicates, minimum of three technical replicates. (C) Immunofluorescence on day 30 comparing the number of DAPI/Islet 1/2 double-positive cells. Statistical analysis: unpaired *t*-test, no significant differences were observed ($p > 0.05$). Data were shown as mean ± SEM; $N = 3$ biological replicates, minimum of three technical replicates.

A

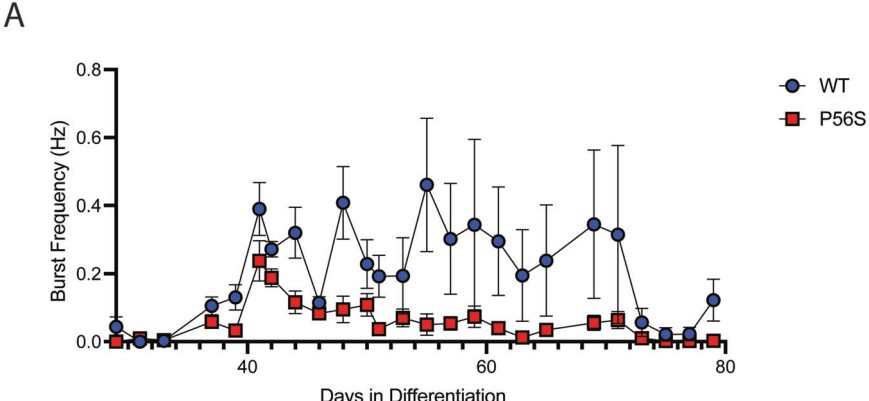

**Figure EV3. Burst frequency of VAPB P56S iPSC-derived motor neurons across differentiation.**

(A) Data presented as mean ± SEM; $N = 8$ technical replicates.

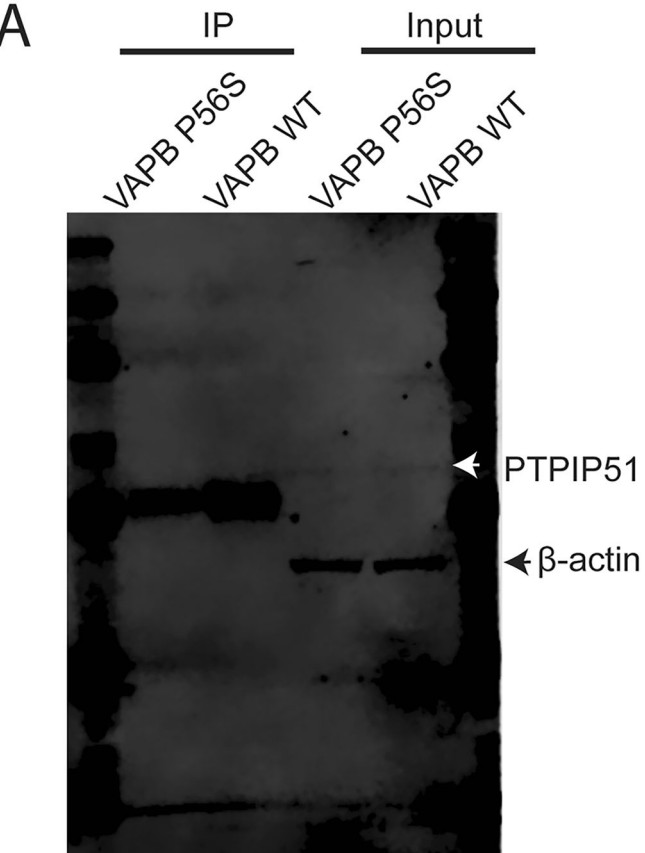

**Figure EV4. Overexposed VAPB co-immunoprecipitation western blot.**

(A) Overexposed western blot showing the PTPIP51 band in the input lane and absence of β-actin in the IP lane.

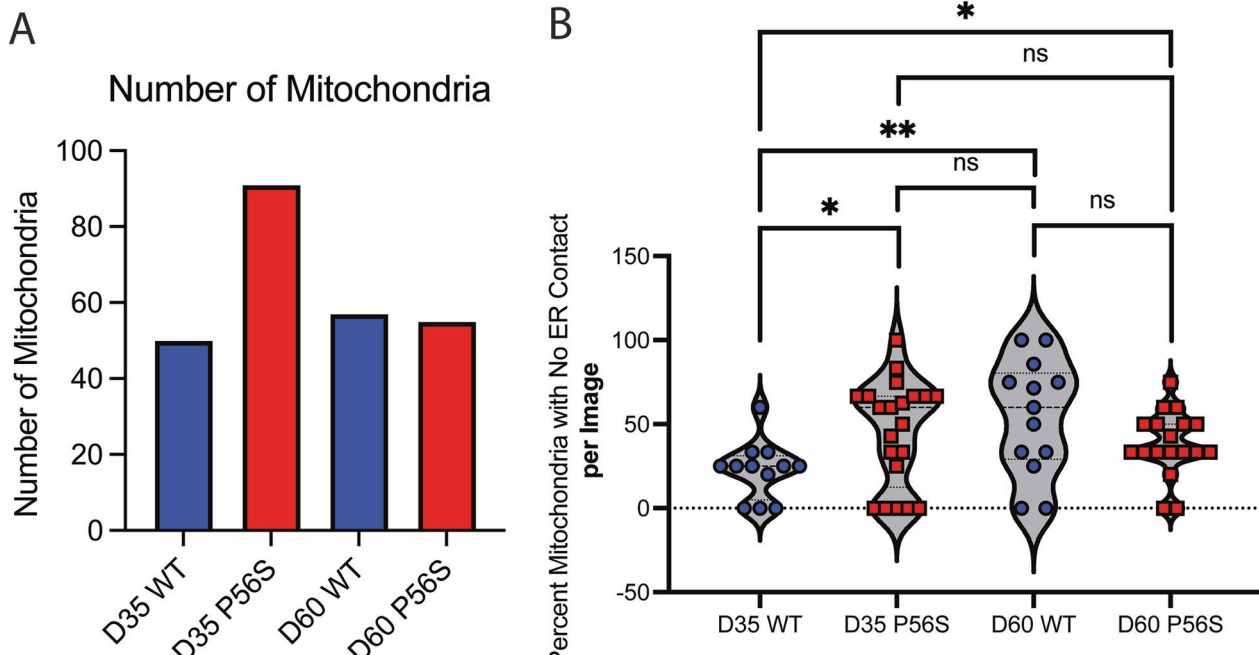

◀ **Figure EV5.  Expanded quantification of ER–mitochondria contacts (MAMs) from electron microscopy.**

(**A**) Number of mitochondria counted for each line across all images taken at time point (day 35 or day 60) from 1 experiment. (**B**) Percent of mitochondria present in each image with no ER-Mitochondrial contact observable. Statistical analysis: unpaired *t*-test: VAPB WT Day 35 vs VAPB P56S Day 35 $p = 0.0248$*, VAPB WT Day 35 vs VAPB WT Day 60 $p = 0.0078$**, VAPB WT Day 35 vs VAPB P56S Day 60 $p = 0.0293$*, Violin plot with median, first and third quartiles denoted as dashed lines, $N = 37$ mitochondria from 13 unique images for D35 WT, 42 mitochondria from 21 unique images for D35 P56S, 24 mitochondria from 13 unique images for D60 WT, 31 mitochondria for 18 unique images for D60 P56S. (**C**) Area of mitochondria in pixels. Statistical analysis: unpaired *t*-test: all *p* values >0.05. Violin plot with median, first and third quartiles denoted as dashed lines, $N = 50$ mitochondria from 13 unique images for D35 WT, 91 mitochondria from 21 unique images for D35 P56S, 57 mitochondria from 13 unique images for D60 WT, 55 mitochondria from 18 unique images for D60 P56S.

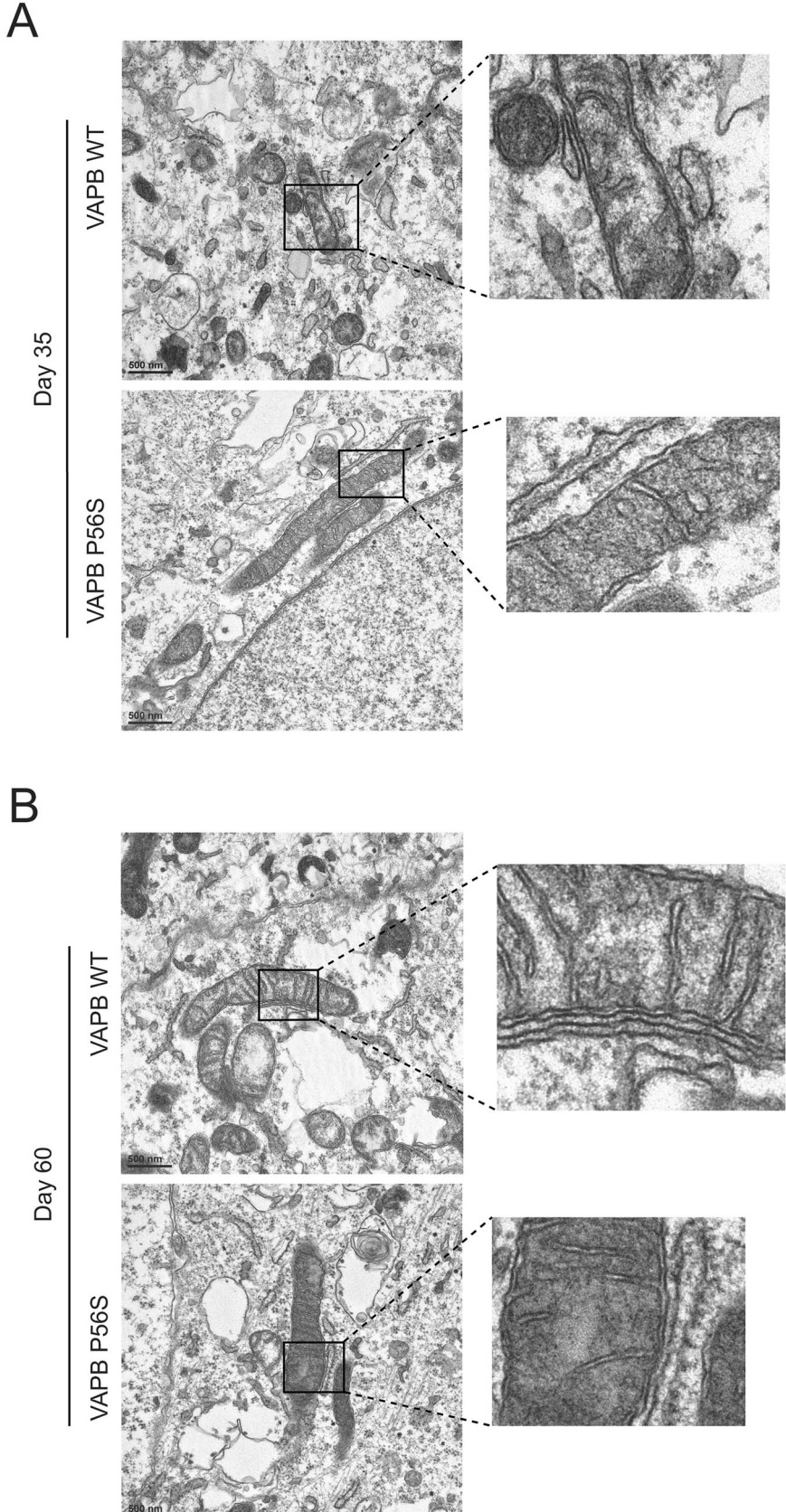

**Figure EV6. Additional electron micrographs of ER–mitochondria contacts during motor neuron differentiation.**

(A) Representative images from day 35 VAPB WT and VAPB P56S motor neurons, with insert zooms highlighting ER–mitochondria contacts. (B) Representative images from day 60 VAPB WT and VAPB P56S motor neurons, with insert zooms as above. Source data are available online for this figure.

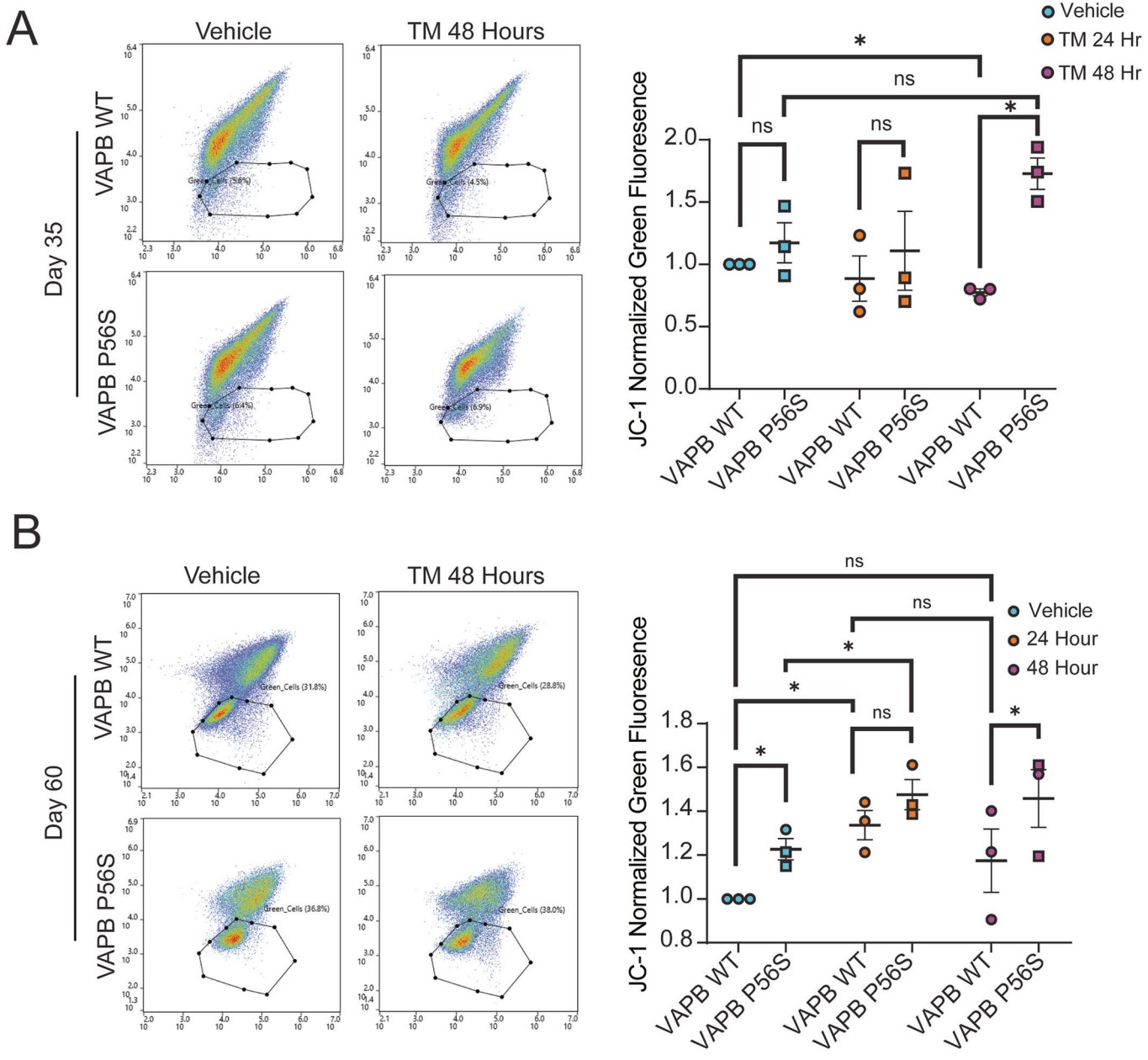

**Figure EV7. VAPB P56S impairs cellular response to stress.**

(**A**) JC-1 staining of day 35 motor neurons after 24 or 48 h of tunicamycin treatment, or untreated control. Data were presented as mean ± SEM; *N* = 3 biological replicates. Statistical analysis: paired *t*-test. Significant comparisons: VAPB WT Vehicle vs VAPB WT TM 48 h (*p* = 0.0141), VAPB WT TM 48 h vs VAPB P56S TM 48 h (*p* = 0.0450). (**B**) Same analysis as (**A**) on day 60. Statistical analysis: paired *t*-test. Significant comparisons: VAPB WT Vehicle vs VAPB P56S Vehicle (*p* = 0.043), VAPB WT Vehicle vs VAPB WT TM 24 h (*p* = 0.0374), VAPB P56S Vehicle vs VAPB P56S TM 24 h (*p* = 0.0229), VAPB WT TM 48 h vs VAPB P56S TM 48 h (*p* = 0.0212).

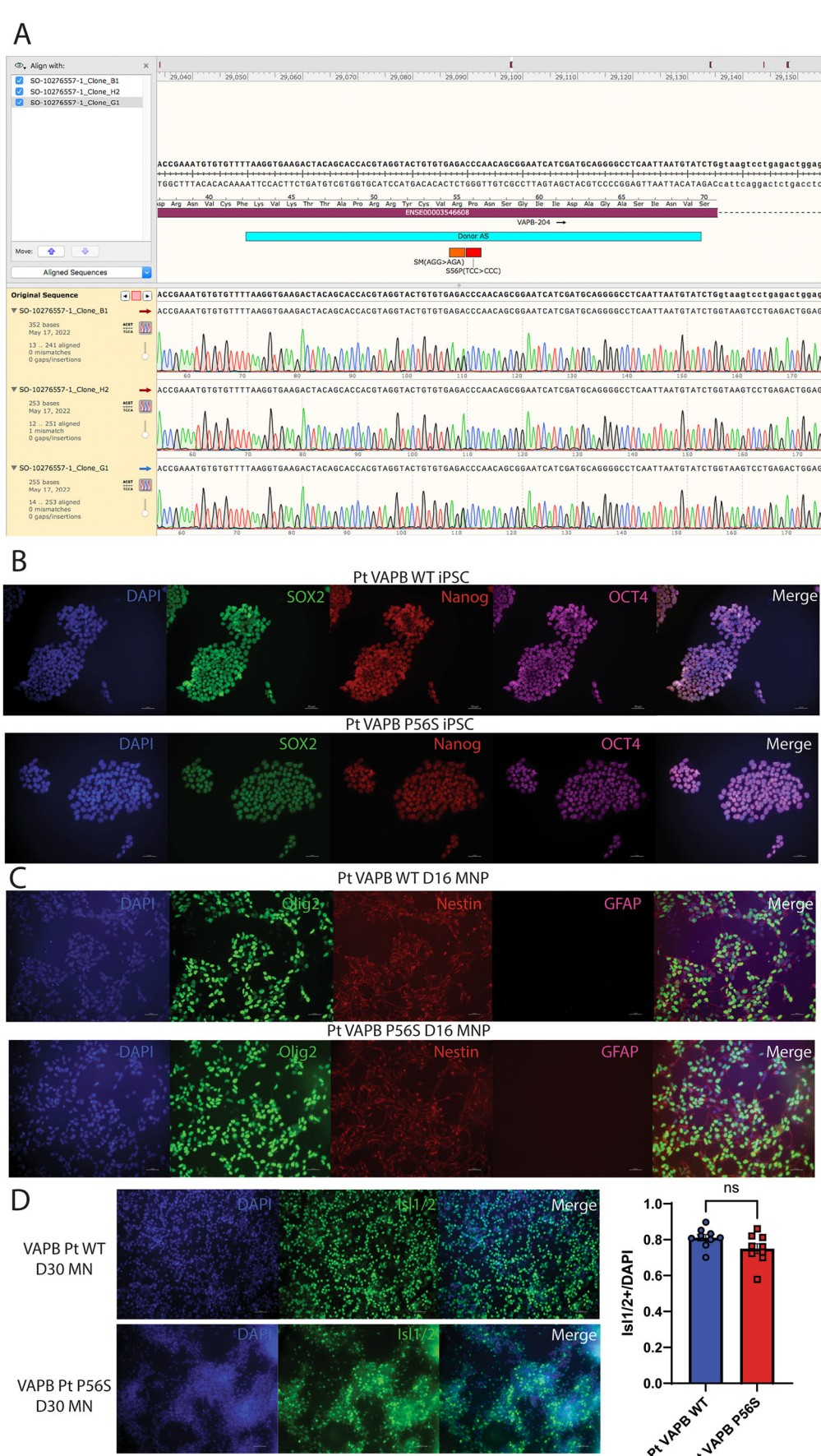

◀ **Figure EV8. Characterization of VAPB patient iPSCs and isogenic control lines.**

(A) Sanger sequencing of the patient iPSC line following CRISPR-Cas9-mediated mutagenesis. The heterozygous P56S mutation (TCC, encoding Ser) was corrected to the WT codon (CCC, encoding Pro). A synonymous mutation (SM) was also introduced at the adjacent PAM site (AGG to AGA, both encoding Arg) via the donor template to prevent re-cutting by Cas9. (B) Immunofluorescence confirming expression of pluripotency markers (SOX2, Nanog, and OCT4) in isogenic iPSC lines. $N = 3$ biological replicates, minimum of three technical replicates. (C) Immunofluorescence on day 15 showing expression of motor neuron progenitor markers Nestin and Olig2, with no GFAP expression. $N = 3$ biological replicates, minimum of three technical replicates. (D) Immunofluorescence on day 30 comparing the number of DAPI/Islet 1/2 double-positive cells. Statistical analysis: unpaired $t$-test, no significant difference observed ($p > 0.05$). Mean ± SEM; $N = 3$ biological replicates, three technical replicates of each biological replicate.

