## [Peer Review File · EMBO Molecular Medicine]

Convergent Activation of the Integrated Stress Response and ER-Mitochondria Uncoupling in VAPB-Associated ALS

Curran Landry, James Costanzo, Miguel Mitne-Neto, Mayana Zatz, Ashleigh Schaffer, Maria Hatzoglou, Alysson Muotri, and Helen Miranda

Corresponding author(s): Helen Miranda (hcm34@case.edu)

Review Timeline:

Transferred from Review Commons:	6th May 25
Editorial Decision:	17th Jun 25
Revision Received:	1st Jul 25
Accepted:	11th Jul 25

Editor: Jingyi Hou

Transaction Report:

This manuscript was transferred to EMBO Molecular Medicine following peer review at Review Commons.

Review #1**1. Evidence, reproducibility and clarity:****Evidence, reproducibility and clarity (Required)**

Landry et al. present characterization of iPSC-derived neurons that inducibly express either WT VAPB or P56S VAPB in the context of a VAPB knockout. They do this by first generating a novel iPSC line with a frameshift knockout in a VAPB, and then selecting lentiviral-transduced clones that express either WT or P56S VAPB from an inducible promoter. The resulting lines are then differentiated using conventional protocols, VAPB expression is induced, and the cells are subjected to a battery of cell biological tests to examine mitochondrial function.

****Major Points:****

1. The method of knocking out and selecting an inducible line is problematic. VAPB is an essential gene-patients with P56S are always heterozygotes, since nonfunctional VAPB is embryonic lethal. Selecting a knockout cell line is already choosing a parent that is very far from physiological, and the reexpression of P56S VAPB as the sole form also is not a good a model for understanding the contributions of P56S to disease. This approach is unusual, as it seems to overlook the advantages of working with iPSCs and patient-derived neurons. Unfortunately, the value of this amazing and rare system is diminished by the design of the selection method.
2. The interactome analysis is not controlled properly to interpret. It is not the total amount of VAPB that needs to be used as the normalization control, since it is already known 90+% of the P56S VAPB is in cytoplasmic aggregates. The authors need to normalize to the amount of VAPB that made it to the contact sites-a much smaller amount in the cells expressing the diseased form. For example, the fact that the authors can still pull down detectable PTPIP51 in Fig. 2e actually argues for the opposite conclusion than what the authors have stated-if the authors have actually expressed just P56S in a true knock out condition, this means that P56S CAN still bind to PTPIP51 (and possibly even better than WT, as several previous papers have suggested-since there is ~100-fold less available for binding). Without an appropriate normalization, the authors cannot make any conclusion about how to interpret the results of this part of the paper.
3. The electron microscopy data is not interpretable in this form. The authors have provided no data at all on how analysis was performed, how contact sites were defined, how samples were collected and ensured to be representative, blinding that was performed,

how sources of bias were accounted for, etc. It is clear even from what little is shown that the authors are not focused on what matters to address their own questions. For example, apart from the P56S Day 35 example, none of the "contact sites" selected for the figure are even possible to be mediated by VAPB, since the distance between the ER and the mitochondria is too far for the maximum tethering distance of VAPB-PTPIP51. Since the authors have neglected to include scale bars in their zooms, the reader cannot be sure of the distance, but it is clearly in excess of 50 nm since there are obviously visible ribosomes between the two organelles. Additionally, the authors provide no information on what "% mitochondria in contact with ER" means (By organelle? By unit surface area? Is the data grouped by cell or all comes from a single cell? How do you account for contact sites vs. proximity by crowding? Etc.).

4. The strange pooling of data without explanation, unusual sample sizes, and lack of clarity about statistical testing, false hypothesis testing, and really any clear rigor in statistics of any kind make it impossible for a reader to have any confidence in the results presented here. The fact that every experiment in the paper has just enough n to trigger statistical significance as determined by the authors raises some concerns, suggesting potential biases. The reliability of these conclusions is questionable, especially if the authors were blinded to the identity of their own samples. This is particularly relevant for the EM data, where the determination of contact sites appears to have been made subjectively.

****Minor points:****

1. It is not accurate to describe Day 60 neurons as "aged" in the context of P56S-induced disease or imply they are a model for human aging. I could be mistaking, as I am not an iPSC expert, but I believe the field uses these terms in the context of iPSC-derived neurons to mean something more akin to "mature". The authors try to invoke this to argue for the relevance of their results to patient disease, unless the authors know this is somehow actually representative of neurons from older patients, I think this is misleading.

2. The JC-1 experiment is not being appropriately controlled. These results are predicted by increased cell or mitochondrial death even if the membrane potentials are identical. The authors need to control for apoptotic signaling if they want to make this conclusion. There is an accepted standard in the mitochondrial field for assaying mitochondrial membrane potential (generally using TMRE or TMRM, but JC-1 can be used with proper controls), but it requires lots of careful controls not performed here (normalization to oligomycin- and FCCP-treated cells as a bare minimum).

3. The flow cytometry experiments are problematic in general since the authors state that

part of their incentive for studying mitochondria in this model is due to effects at synapses, and the sample preparation for the cytometer involved dissociating the cells (i.e.-removing all of the processes where synapses mostly reside).

4. The normalization for VAPB in the inducible lines is unclear-how is normalization performed simultaneously to two genes at once? The authors do not provide enough information for us to understand what they have actually done, and I wonder if the data presented in the supplement on this is actually sufficiently different from random noise to be interpretable, since no statistics of any kind are given.

5. I don't think the tunicamycin experiments make sense in this context. The authors start with premise that I do not understand: "if the decrease in MERC was underlying the decrease in MMP seen later in differentiation, inducing cell stress early in differentiation could mimic the decreased MMP." Most cell stress pathways enhance ER-mito contact, not decrease it, so I am not sure why they expected this to work this way. They then continue: "We selected tunicamycin, an ER stressor, as VAPB is an ER protein, and if the decreased MMP could be caused, at least partially, by loss of MERCs, ER stress would likely exacerbate it." I don't understand this either- Tunicamycin is not a general ER-stressing agent-it is a specific inhibitor of some N-linked glycosylation-maturation pathways in the ER lumen, which causes ER stress by dysregulation of misfolded protein pathways. Since VAPB has no luminal domains to speak of, is not known to interact with the protein folding and maturation machinery at all, and Tunicamycin has no obvious connection I'm aware of to MERCs, I am not able to follow the authors' intentions or conclusions here. I suspect this needs a major rewrite to explain what the goals were and how the authors controlled for their findings.

2. Significance:

Significance (Required)

While the idea of assaying the function of ALS-causing VAPB mutants in iPSC-derived neurons is great and would be a great asset to the field, the execution here raises significant concerns. It is difficult to draw clear conclusions from the presented data. Necessary controls are either incorrectly applied or missing, the methods section lacks crucial details for reproducibility, and the figures suggest a lack of appropriate blinding, with cherry-picking evident even in the "representative" images. There are also major issues with the entire premise of how the lines were generated, since VAPB knockout cells are highly aberrant lines, the authors have likely selected for all sorts of mitochondrial pathways that would not be operating in an actual patient neuron.

Claims about mitochondrial dysfunction could potentially mislead the field, as such

conclusions do not seem to be supported by the actual data. To be suitable for publication, the study needs substantial revisions, including proper controls, blinding, and detailed methodological information for reproducibility. I understand the challenges and costs associated with using iPSC-derived neurons, but focusing on a few well-controlled experiments would be far more beneficial than presenting numerous, less interpretable findings.

3. How much time do you estimate the authors will need to complete the suggested revisions:

Estimated time to Complete Revisions (Required)

(Decision Recommendation)

More than 6 months

4. Review Commons values the work of reviewers and encourages them to get credit for their work. Select 'Yes' below to register your reviewing activity at Web of Science Reviewer Recognition Service (formerly Publons); note that the content of your review will not be visible on Web of Science.

No

Review #2

1. Evidence, reproducibility and clarity:

Evidence, reproducibility and clarity (Required)

Mutations in the VAPB gene are a cause of amyotrophic lateral sclerosis (ALS), a human motor neuron disease. To define the mechanisms by which mutations in VAPB cause motor neuron degeneration, the authors establish a new human iPSC-derived motor neuron model. They start by using CRISPR to knockout the VAPB gene and then introduce a lentivirus encoding a doxycycline-inducible construct to express WT or mutant VAPB. They then phenotypically characterize these WT and mutant motor neurons including using multi-electrode array (MEA), which revealed neuronal firing deficits in mutant motor neurons. They performed protein interaction studies WT vs mutant VAPB motor neuron and identified decreased binding to PTPIP51 in the mutant VAPB motor neurons.

Phenotypically, the authors report that the VAPB mutant motor neurons exhibit decreased

mitochondria / ER contacts (MERC) in mutant motor neurons compared to WT as well as decreased mitochondrial membrane potential. They report that these mitochondrial defects lead to heightened sensitivity to ER stress and activation of the integrated stress response, which could be rescued by treatment with ISRIB. Importantly, the neuronal firing defects are also rescued by ISRIB, providing compelling evidence that these defects are tied to activation of ER stress.

Overall, this paper presents novel functional analyses of an important ALS gene, VAPB in disease-relevant cell types (human motor neurons). I have the following comments and suggestions for the authors to consider.

1. Why did the authors decide to make VAPB knockouts and then introduce the WT or P56S VAPB constructs on a lentivirus instead of generating the point mutations (or correcting them) directly in the endogenous locus? Data in Extended Fig. 1c and Extended Fig. 2a indicate significant differences in either the kinetics of WT vs. P56S VAPB expression (1c) or levels (2a). It seems important to be able to compare comparable levels of WT and mutant proteins, especially for the interpretation of the subsequent IP-MS experiments to identify PTP151. The authors may wish to consider generating (or obtaining) isogenic lines harboring the mutations at the endogenous locus so that equal levels of expression of WT and mutant VAPB can be assessed.

2. The authors highlight PTP151 binding to VAPB as a way to promote mitochondria ER contacts (MERC). They provide evidence that this association is diminished by the P56S VAPB mutation. This raises an important question. How does PTP151 binding connect with other phenotypes, such as the neuronal firing and ER stress sensitivity? Can the authors consider experiments to test this directly? For example, is there a way to drive PTP151 : VAPB interactions even in the face of mutant VAPB and see if this rescues the MERC defects and other phenotypes?

3. The authors propose that the detachment of the mitochondria from the ER most likely be the cause for why their mutant motor neurons are more sensitive to ER stressors. Along the lines of the above, is there a way to test this hypothesis directly? Can they use other means to promote ER mitochondria association even in the face of VAPB mutation and test if this rescues phenotypes?

****Referee Cross-commenting****

There seems to be concurrence between Reviewer 1 and 2 about the interest in the VAPB gene but that the specific approaches and analyses methods used to study mutations in

this gene (knockout and then over expression of WT and mutant version) are not a faithful representation of the in vivo situation (heterozygous mutations) and both provide suggestions for improvement of the study design.

****Editorial Note****

This Editorial Note by the Review Commons editorial team was communicated to the author in response to their request for clarification and contextualization of the referee report of reviewer #1.

Since reviewer #1 did not clarify what was requested by the editorial office, we included the present Editorial Note in the review process after re-analyzing the manuscript in detail again and the referee report of reviewer #1.

We agree with the authors that the wording used by reviewer #1 is problematic. However, we also see that the substance of the points raised by this reviewer is relevant and affects the study's conclusions. Below, we have included our comments on the individual points and quotes highlighted in your letter.

Reviewer #1: "The strange pooling of data without explanation"

- When looking into the figures and their captions in more detail, we could also not understand the nature of the replicates and how the data was aggregated or "pooled". In Figure 1, the stated number of replicates is "N=8 separate wells". It is unclear whether these are 8 wells from a single dissociation/replating procedure (the procedure is described in Materials & Methods as follows: "Motor neurons were dissociated on day 25 of differentiation and re-plated onto 48-well MEA plate") or whether the eight are sampled across multiple plates across cultures obtained from independent dissociations procedures.

- In Figure 3, the number of replicates is "N=13-21 images". Here, it is unclear whether these images come from the same or independent samples, how many quantifications were performed per image, and how many images per sample were used.

- We also note that replicates are not mentioned in the proteomics analysis.

Reviewer #1: "unusual sample sizes":

- The wording is indeed not very explicit, but we believe it is reasonable to assume that this point refers to "N=13-21 images" and that it is not clear how the data were pooled. The reviewer makes the related point: "Is the data grouped by cell or all comes from a single cell?", which provides further context to this point.

"lack of clarity about statistical testing":

- We agree that without a clear description of the nature of the replicates, the statistical analysis is unclear.

"false hypothesis testing":

- We agree with the authors that the reviewer is unclear.

"The fact that every experiment in the paper has just enough n to trigger statistical significance as determined by the authors raises some concerns, suggesting potential biases."

- We agree that this is an inappropriate statement in absence of evidence or detailed argumentation; we very much regret not having caught this statement up front.

"The reliability of these conclusions is questionable, especially if the authors were blinded to the identity of their own samples.":

- This is a typo; the word "not" is missing. It should read: "if the authors were NOT blinded to the identity..." and refers to concerns raised by the reviewers about evaluating the EM images.

"The figures suggest a lack of appropriate blinding, with cherry-picking evident even in the 'representative' images"

- We agree the wording is somewhat problematic. However, we also feel that there is a discrepancy between the differences highlighted between the EM images shown in Fig 3A and a rather modest change of the median by only a few percent, as shown in the respective violin plots. We agree with the reviewer that the images of Fig 3A might, therefore, not be "representative" of the quantified changes.

We agree that there are statements in this review that are written in a style and tone that is not appropriate. We greatly apologize for this and, we should have caught these issues beforehand.

At the same time, this reviewer raises significant issues about the study. In this case, we cannot eliminate the entire review since the points raised are relevant to the conclusiveness of the study.

2. Significance:

Significance (Required)

The new iPSC-derived system to study VAPB mutations in human motor neurons is significant and has led the authors to discover new functions for VAPB (i.e., mitochondria-ER contacts). The significance and impact of the study, in my opinion, would be increased if the authors considered using motor neuron lines expressing comparable levels of WT and mutant VAPB, preferably from the endogenous location under physiological conditions. Their discovery of a role of defective mitochondria-ER contact as making VAPB mutant motor neurons more sensitive to ER stress would be bolstered by experiments to directly test this hypothesis by rescuing the contact defects.

3. How much time do you estimate the authors will need to complete the suggested revisions:

Estimated time to Complete Revisions (Required)

(Decision Recommendation)

More than 6 months

4. Review Commons values the work of reviewers and encourages them to get credit for their work. Select 'Yes' below to register your reviewing activity at Web of Science Reviewer Recognition Service (formerly Publons); note that the content of your review will not be visible on Web of Science.

No

Full Revision

Manuscript number: RC-2024-02520

Corresponding author(s): Helen Miranda

[Please use this template only if the submitted manuscript should be considered by the affiliate journal as a full revision in response to the points raised by the reviewers.]

*If you wish to submit a preliminary revision with a revision plan, please use our "Revision Plan" template. **It is important to use the appropriate template to clearly inform the editors of your intentions.**]*

1. General Statements [optional]

****Referee Cross-commenting****

There seems to be concurrence between Reviewer 1 and 2 about the interest in the VAPB gene but that the specific approaches and analyses methods used to study mutations in this gene (knockout and then over expression of WT and mutant version) are not a faithful representation of the in vivo situation (heterozygous mutations) and both provide suggestions for improvement of the study design.

Manuscript New Title: **“Convergent Activation of the Integrated Stress Response and ER–Mitochondria Uncoupling in VAPB-Associated ALS”**

We thank the reviewers for their thoughtful evaluation and for recognizing the contribution of our study to the understanding of VAPB biology and ALS pathogenesis. We especially appreciate the reviewers' positive remarks regarding the significance of our iPSC-derived motor neuron model and its potential to advance the study of ALS mechanisms—*“the idea of assaying the function of ALS-causing VAPB mutants in iPSC-derived neurons is great and would be a great asset to the field”* (Reviewer 1), and *“the new iPSC-derived system to study VAPB mutations in human motor neurons is significant and has led the authors to discover new functions for VAPB (i.e., mitochondria-ER contacts)”* (Reviewer 2).

In this work, we identify the **first mechanistic link between the ALS-associated VAPB P56S mutation and activation of the Integrated Stress Response (ISR)** via mitochondrial dysfunction and ER–mitochondria uncoupling in human motor neurons. This genotype-to-mechanism insight helps explain how a specific familial mutation drives ISR-mediated neurodegeneration, with broader implications for ALS biology.

Importantly, our findings offer a possible explanation for **the failure of recent ISR-targeting clinical trials**, which did not account for patient genotype. By revealing a mutation-specific

mechanism of ISR activation, our work provides **a rationale for patient stratification in future ISR-based ALS therapies.**

Both reviewers raised concerns regarding the physiological relevance of our original doxycycline-inducible VAPB system. While that model was designed to enable mechanistic dissection of mutant-specific effects, we fully agree that patient-derived validation is critical. To this end, we have reprogrammed ALS8 patient fibroblasts and generated isogenic CRISPR-corrected controls. These new data confirm that the **ISR activation, mitochondrial dysfunction, and hypotranslation identified in the inducible system are also present in patient-derived motor neurons, and are rescued by ISRIB.** This validation reinforces both the **physiological relevance** and **translational significance** of our findings.

All other reviewer critiques—regarding proteomics normalization, EM quantification, statistical transparency, and more—have been addressed in detail and are fully resolved in this revised submission. The only reviewer suggestion not addressed experimentally was the artificial tethering of ER and mitochondria, for which we explain the **technical and mechanistic confounds** in the final section of the letter.

We believe that this revised manuscript now presents a **robust, mutation-specific mechanism of ALS pathogenesis**, and supports the development of **genotype-informed therapeutic strategies** across both familial and potentially sporadic forms of ALS.

Reviewer #1 (Evidence, reproducibility and clarity (Required)):

Landry et al. present characterization of iPSC-derived neurons that inducibly express either WT VAPB or P56S VAPB in the context of a VAPB knockout. They do this by first generating a novel iPSC line with a frameshift knockout in a VAPB, and then selecting lentiviral-transduced clones that express either WT or P56S VAPB from an inducible promoter. The resulting lines are then differentiated using conventional protocols, VAPB expression is induced, and the cells are subjected to a battery of cell biological tests to examine mitochondrial function.

Major Points:

Reviewer #1 Major Point 1. The method of knocking out and selecting an inducible line is problematic. VAPB is an essential gene—patients with P56S are always heterozygotes, since nonfunctional VAPB is embryonic lethal. Selecting a knockout cell line is already choosing a parent that is very far from physiological, and the reexpression of P56S VAPB as the sole form also is not a good model for understanding the contributions of P56S to disease. This approach is unusual, as it seems to overlook the advantages of working with iPSCs and patient-derived neurons. Unfortunately, the value of this amazing and rare system is diminished by the design of the selection method.

Revisions

The development of the inducible system for VAPB was specifically designed to enable a systematic investigation of the effects of mutant VAPB (VAPB P56S) on cellular homeostasis while minimizing confounding influences from the wild-type (WT) protein. Additionally, this system allowed us to assess VAPB P56S binding partners and compare them to those of VAPB WT, which would not have been feasible in the context of heterozygous ALS8 patient cells.

Nonetheless, we acknowledge the significance of studying ALS patient-derived iPSCs. To address this, we obtained fibroblasts from an ALS8 patient carrying the heterozygous VAPB P56S mutation, originating from a genetic background distinct from the cells used in our inducible system. These fibroblasts were reprogrammed into iPSCs in our laboratory, followed by CRISPR/Cas9-mediated genome editing to generate isogenic corrected iPSCs as controls.

The resulting iPSC isogenic pair was differentiated into motor neurons following the protocol described in our manuscript (Fig EV8). Notably, ALS8 patient iPSC-derived motor neurons exhibited reduced mRNA translation, as assessed by the SUnSET assay (Fig. 6A), along with a decrease in mitochondrial membrane potential, as determined using the JC-1 assay (Fig. 6B). These findings confirm that the hypotranslation and mitochondrial dysfunction initially identified in VAPB P56S doxycycline-inducible iPSC-derived motor neurons were successfully recapitulated in ALS8 patient iPSC-derived motor neurons. Furthermore, ISRIB treatment effectively rescued these phenotypic defects.

Overall, these results demonstrate that the molecular and cellular abnormalities identified in the original inducible system can be reliably reproduced in an ALS patient-derived model with a different genetic background, thereby reinforcing the significance and broader applicability of our findings.

Reviewer #1 Major Point 2. The interactome analysis is not controlled properly to interpret. It is not the total amount of VAPB that needs to be used as the normalization control, since it is already known 90+% of the P56S VAPB is in cytoplasmic aggregates. The authors need to normalize to the amount of VAPB that made it to the contact sites-a much smaller amount in the cells expressing the diseased form. For example, the fact that the authors can still pull down detectable PTPIP51 in Fig. 2e actually argues for the opposite conclusion than what the authors have stated-if the authors have actually expressed just P56S in a true knock out condition, this means that P56S CAN still bind to PTPIP51 (and possibly even better than WT, as several previous papers have suggested-since there is ~100-fold less available for binding). Without an appropriate normalization, the authors cannot make any conclusion about how to interpret the results of this part of the paper.

Revisions

We sincerely thank Reviewer 1 for highlighting this critical point. Previous studies have demonstrated that the VAPB P56S mutation increases its binding affinity for PTPIP51; however,

it has been proposed that the overall reduction in VAPB levels in cells harboring the P56S mutation leads to a decrease in ER-mitochondrial contacts despite the enhanced affinity (De Vos et al., 2012).

To address this, we have repeated the co-immunoprecipitation experiment and normalized the data to VAPB levels. Consistent with Reviewer 1's hypothesis, when normalized to soluble VAPB, we observe an increased affinity of VAPB P56S for PTPIP51. However, the total amount of PTPIP51 co-immunoprecipitated with VAPB remains significantly lower in the mutant compared to WT, likely due to the overall reduced levels of soluble VAPB P56S. This finding aligns with both Reviewer 1's comment and the previous observations reported by De Vos et al. (2012).

Figure 2E has been updated to reflect the normalized co-immunoprecipitation data.

Citation:

1. De Vos, K. J. *et al.* VAPB interacts with the mitochondrial protein PTPIP51 to regulate calcium homeostasis. *Hum Mol Genet* **21**, 1299-1311, doi:10.1093/hmg/ddr559 (2012).

Reviewer #1 Major Point 3. The electron microscopy data is not interpretable in this form. The authors have provided no data at all on how analysis was performed, how contact sites were defined, how samples were collected and ensured to be representative, blinding that was performed, how sources of bias were accounted for, etc. It is clear even from what little is shown that the authors are not focused on what matters to address their own questions. For example, apart from the P56S Day 35 example, none of the "contact sites" selected for the figure are even possible to be mediated by VAPB, since the distance between the ER and the mitochondria is too far for the maximum tethering distance of VAPB-PTPIP51. Since the authors have neglected to include scale bars in their zooms, the reader cannot be sure of the distance, but it is clearly in excess of 50 nm since there are obviously visible ribosomes between the two organelles. Additionally, the authors provide no information on what "% mitochondria in contact with ER" means (By organelle? By unit surface area? Is the data grouped by cell or all comes from a single cell? How do you account for contact sites vs. proximity by crowding? Etc.).

Revisions

We thank Reviewer 1 for their insightful comments on the analysis of the electron microscopy (EM) data and recognize the need for greater clarity in describing our quantification approach. To address this, we have revised the Electron Microscopy section of the Methods to explicitly detail our methodology for quantifying ER-mitochondria-associated membranes (ER-MAMs), as follows:

"A series of images at various magnifications were provided, and data were collected from unique images at the highest magnification for each condition: D35 WT (13 unique images), D35 P56S (21 unique images), D60 WT (13 unique images), and D60 P56S (18 unique images). All images for a given condition originated from a single well of a 12 mm Snapwell™ Insert with 0.4 µm Pore Polyester Membranes (Corning). No information on cell grouping or

sampling strategy was supplied with the images; therefore, we treated the dataset as a random sampling of the culture. Images were blinded, and quantification was performed using FIJI. Mitochondria were identified based on the presence of cristae and a double membrane. The mitochondrial perimeter was traced using the free-draw tool, and the length of ER membranes within 50 nm of this perimeter was quantified. The final measurement represents the percentage of each mitochondrion's perimeter in contact with the ER, aggregating all visually distinct ER-MAMs, as continuity beyond the imaging plane cannot be determined (Cosson et al., 2012; Csordás et al., 2010; Stoica et al., 2014). Each data point on the graph corresponds to a single mitochondrion, with data collected from multiple cells across the unique images provided by the Core, originating from a single biological replicate."

Regarding the quantification of ER-MAM distances, VAPB has not been definitively localized exclusively to either the rough or smooth ER. To ensure comprehensive analysis, we quantified ER-MAMs without restricting our assessment to a specific ER subdomain. We adopted a 50 nm threshold as the maximum distance for defining ER-MAMs, a well-established criterion that Reviewer 1 also referenced.

Furthermore, we disagree with Reviewer 1's assertion that the presence of ribosomes should justify extending the ER-MAM threshold beyond 50 nm. Ribosomes in human cells have a well-documented diameter of 20–30 nm (Anger et al., 2013), which does not support the claim that an observed ribosome within the contact site necessitates a redefinition of the ER-MAM boundary.

We stand by our methodological approach and have updated the manuscript to ensure precision and clarity in our EM data analysis.

Citations:

1. Cosson, P., Marchetti, A., Ravazzola, M. & Orci, L. Mitofusin-2 independent juxtaposition of endoplasmic reticulum and mitochondria: an ultrastructural study. *PLoS One* **7**, e46293 (2012).
2. Csordás, G. *et al.* Imaging interorganelle contacts and local calcium dynamics at the ER-mitochondrial interface. *Mol Cell* **39**, 121-132 (2010).
3. Stoica, R. *et al.* ER-mitochondria associations are regulated by the VAPB-PTPIP51 interaction and are disrupted by ALS/FTD-associated TDP-43. *Nat Commun* **5**, 3996 (2014).
4. Anger AM, Armache JP, Berninghausen O, Habeck M, Subklewe M, Wilson DN, Beckmann R. Structures of the human and Drosophila 80S ribosome. *Nature*. 2013 May 2;497(7447):80-5. doi: 10.1038/nature12104. PMID: 23636399.

We would like to thank the Editor of Review Commons for clarifying Reviewer #1's Major Point 4 and will be responding to the Editor's interpretations as detailed in the Editorial Note.

Reviewer #1 Major Point 4. The strange pooling of data without explanation, unusual sample sizes, and lack of clarity about statistical testing, false hypothesis testing, and really any clear rigor in statistics of any kind make it impossible for a reader to have any confidence in the results presented here. The fact that every experiment in the paper has just enough n to trigger statistical significance as determined by the authors raises some concerns, suggesting potential

biases. The reliability of these conclusions is questionable, especially if the authors were blinded to the identity of their own samples. This is particularly relevant for the EM data, where the determination of contact sites appears to have been made subjectively.

Reviewer #1: "The strange pooling of data without explanation"

Editorial Note- *When looking into the figures and their captions in more detail, we could also not understand the nature of the replicates and how the data was aggregated or "pooled". In Figure 1, the stated number of replicates is "N=8 separate wells". It is unclear whether these are 8 wells from a single dissociation/replating procedure (the procedure is described in Materials & Methods as follows: "Motor neurons were dissociated on day 25 of differentiation and re-plated onto 48-well MEA plate") or whether the eight are sampled across multiple plates across cultures obtained from independent dissociations procedures.*

- We apologize for the lack of clarity and specificity. We have updated the Multi-Electrode Array Recordings portion of the Methods Section with the following:
"iPSC-derived MNs from a single well of a 6-well plate thawed as day 15 MNP were dissociated and plated across 8 wells of the MEA plate. Each point on the graph is an average of the weighted mean firing rate of those 8 wells, normalized for cell count across genotypes, obtained after all firings were recorded by dissociating 2 wells per line, counting and averaging the cell numbers, and then normalizing all firings by the ratio of cell number between WT and P56S. Wells with no firing detected were excluded from quantification."

Editorial Note- *In Figure 3, the number of replicates is "N=13-21 images". Here, it is unclear whether these images come from the same or independent samples, how many quantifications were performed per image, and how many images per sample were used.*

- We have updated the Electron Microscopy Methods Section with the following:
"We were provided with a series of images and magnifications and were able to gather data from unique images at the highest magnification level for each of the following categories: D35 WT: 13 unique images, D35 P56S: 21 unique images, D60 WT 13 unique images, D60 P56S: 18 unique images. All images for a given line come from a single well of a 12 mm Snapwell™ Insert with 0.4 μm Pore Polyester Membranes (Corning). No indication of cell grouping or sampling techniques was provided with the images, therefore the images were quantified as a random sampling of the culture. Images were then blinded, and FIJI was used to quantify."
We are happy to make all images publicly available.

Editorial Note- *We also note that replicates are not mentioned in the proteomics analysis.*

- We apologize for missing this and thank the editor for mentioning it. The Proteomics portion of the methods section has been updated with the following:
"The identification of VAPB binding partners via mass spectrometry was performed with one biological sample, while the validation of VAPB-PTPIP51 binding via co-immunoprecipitation and Western Blot was performed with three separate biological replicates."

Full Revision

Reviewer #1: "unusual sample sizes":

Editorial Note- *The wording is indeed not very explicit, but we believe it is reasonable to assume that this point refers to "N=13-21 images" and that it is not clear how the data were pooled. The reviewer makes the related point: "Is the data grouped by cell or all comes from a single cell?", which provides further context to this point.*

- We thank the editor for this clarification, our response to Reviewer #1 Major Point 3 details the updates to Electron Microscopy section of the Methods and covers this. All images were provided to us by the Case Western Reserve University Electron Microscopy Core based on the number of quality images their team were able to obtain from our samples.

Reviewer #1: "lack of clarity about statistical testing":

Editorial Note- *We agree that without a clear description of the nature of the replicates, the statistical analysis is unclear.*

- We hope with the updated clarity on the description of the nature of the replicates as detailed above, the nature of the statistical analysis is clearer. In addition, we have added a Statistical Analysis subsection in the Methods Section.

Reviewer #1: "The reliability of these conclusions is questionable, especially if the authors were blinded to the identity of their own samples.":

Editorial Note- *This is a typo; the word "not" is missing. It should read: "if the authors were NOT blinded to the identity..." and refers to concerns raised by the reviewers about evaluating the EM images.*

- Each unique image was blindly counted.. The images were blinded upon receiving it from the core, and then quantification was performed on the blinded images. The Electron Microscopy portion of the methods section has been updated to make this more clear:
"We were provided with a series of images and magnifications and were able to gather data from unique images at the highest magnification level for each of the following categories: D35 WT: 13 unique images, D35 P56S: 21 unique images, D60 WT 13 unique images, D60 P56S: 18 unique images. All images for a given line come from a single well of a 12 mm Snapwell™ Insert with 0.4 μm Pore Polyester Membranes (Corning). No indication of cell grouping or sampling techniques was provided with the images, therefore the images were quantified as a random sampling of the culture. Images were then blinded, and FIJI was used to quantify."

Reviewer #1: "The figures suggest a lack of appropriate blinding, with cherry-picking evident even in the 'representative' images"

Editorial Note- *We agree the wording is somewhat problematic. However, we also feel that there is a discrepancy between the differences highlighted between the EM images shown in Fig 3A and a rather modest change of the median by only a few percent, as shown in the respective*

violin plots. We agree with the reviewer that the images of Fig 3A might, therefore, not be “representative” of the quantified changes.

- We appreciate the editor's clarification and have selected images that more accurately represent the subtle changes in ER-MAMs observed in our quantification. These images have been included in **Figure EV6** and referenced accordingly in the manuscript to ensure a balanced depiction of our findings. Additionally, we are prepared to make all images publicly available.

We would like to again thank the editor for their clarification on Reviewer #1's Major Point 4 as well as their agreement on the inappropriate nature of some of Reviewer #1's comments.

Minor points

Reviewer#1 Minor point 1. It is not accurate to describe Day 60 neurons as "aged" in the context of P56S-induced disease or imply they are a model for human aging. I could be mistaking, as I am not an iPSC expert, but I believe the field uses these terms in the context of iPSC-derived neurons to mean something more akin to "mature". The authors try to invoke this to argue for the relevance of their results to patient disease, unless the authors know this is somehow actually representative of neurons from older patients, I think this is misleading.

Revisions

We apologize for any confusion. Our use of the term "aged" was intended solely as a relative descriptor, indicating that day 60 motor neurons had been maintained in culture for a longer duration than day 35 motor neurons. It was not meant to suggest that these neurons represent a specific age or disease state, but rather that they had been cultured for an extended period.

Furthermore, we use the term "mature" specifically in the context of motor neuron differentiation to indicate the expression of motor neuron-specific markers, which occurs by day 25 of differentiation. To avoid ambiguity, we have revised the manuscript to use the term "culture time" instead, ensuring clarity in our terminology.

Reviewer #1 Minor Point 2. The JC-1 experiment is not being appropriately controlled. These results are predicted by increased cell or mitochondrial death even if the membrane potentials are identical. The authors need to control for apoptotic signaling if they want to make this conclusion. There is an accepted standard in the mitochondrial field for assaying mitochondrial membrane potential (generally using TMRE or TMRM, but JC-1 can be used with proper controls), but it requires lots of careful controls not performed here (normalization to oligomycin- and FCCP-treated cells as a bare minimum).

Revisions

We would like to thank Reviewer 1 for this comment. We apologize for the omission, and we did treat the cells with CCCP provided in the JC-1 kit as a positive control (Representative data - Figure A). The JC-1 subsection of the methods has been updated to reflect this with the following: “A separate aliquot of cell suspension was also incubated with 4 μ L of the supplied

50mM CCCP for 15 min prior to JC-1 dye addition, to act as a positive control and ensure the JC-1 dye was correctly detecting low MMP populations.”

Figure A: Scatter plot of day 35 motor neurons treated with 200 μ M CCCP for 15 minutes, followed by JC-1 mitochondrial dye staining (see Methods). The X-axis represents red fluorescence, and the Y-axis represents green fluorescence. The gate highlights the population with high green fluorescence.

Reviewer#1 Minor point 3. The flow cytometry experiments are problematic in general since the authors state that part of their incentive for studying mitochondria in this model is due to effects at synapses, and the sample preparation for the cytometer involved dissociating the cells (i.e.- removing all of the processes where synapses mostly reside).

Revisions

We thank Reviewer #1 for this comment. Our citation of the study by Gómez-Suaga et al. (2019) was not intended to suggest that our investigation focuses exclusively on mitochondria at synapses but rather to provide context on the current understanding of the field. To clarify this point, we have revised the manuscript to include the following statement: "*It has also been shown that this interaction can occur at synapses, and disruptions to it may impact synaptic activity (Gómez-Suaga et al., 2019).*"

Citation:

1. Gómez-Suaga, P. et al. The VAPB-PTIP51 endoplasmic reticulum-mitochondria tethering proteins are present in neuronal synapses and regulate synaptic activity. *Acta Neuropathologica Communications* 7, 35, doi:10.1186/s40478-019-0688-4 (2019).

Reviewer#1 Minor point 4. The normalization for VAPB in the inducible lines is unclear-how is normalization performed simultaneously to two genes at once? The authors do not provide enough information for us to understand what they have actually done, and I wonder if the data presented in the supplement on this is actually sufficiently different from random noise to be interpretable, since no statistics of any kind are given.

Revisions

In response, we have added a qPCR section to the Methods, detailing our experimental approach as follows:

"Quantitative PCR: RNA was extracted using TRIzol Reagent (Thermo Fisher), and the procedure was performed according to their provided protocol. cDNA was generated using SuperScript™ IV VIL0™ Master Mix (Thermo Fisher), following the manufacturer's instructions. qPCR was conducted using PowerTrack™ SYBR Green Master Mix for qPCR (Thermo Fisher), following the provided protocol, on a BioRad CFX96 thermocycler. Samples were run in triplicate. Quantification was performed using CFX Maestro software (BioRad). VAPB expression was normalized to Neomycin and RPL3 using the software, and the resultant expression values were graphed along with the provided SEM, per standards in the field (Livak & Schmittgen, 2001; Wong & Medrano, 2005)."

Additionally, we have modified the graph to more clearly illustrate the comparison between VAPB WT and P56S, emphasizing that there is no significant difference in mRNA expression.

Citations

1. Wong, M. L. & Medrano, J. F. Real-time PCR for mRNA quantitation. *Biotechniques* **39**, 75-85 (2005).
2. Livak, K. J. & Schmittgen, T. D. Analysis of relative gene expression data using real-time quantitative PCR and the 2^{(-Delta Delta C(T))} Method. *Methods* **25**, 402-408 (2001).

Reviewer#1 Minor point 5. I don't think the tunicamycin experiments make sense in this context. The authors start with premise that I do not understand: "if the decrease in MERC was underlying the decrease in MMP seen later in differentiation, inducing cell stress early in differentiation could mimic the decreased MMP." Most cell stress pathways enhance ER-mito contact, not decrease it, so I am not sure why they expected this to work this way. They then continue: "We selected tunicamycin, an ER stressor, as VAPB is an ER protein, and if the decreased MMP could be caused, at least partially, by loss of MERCs, ER stress would likely exacerbate it." I don't understand this either- Tunicamycin is not a general ER-stressing agent-it is a specific inhibitor of some N-linked glycosylation-maturation pathways in the ER lumen, which causes ER stress by dysregulation of misfolded protein pathways. Since VAPB has no luminal domains to speak of, is not known to interact with the protein folding and maturation machinery at all, and Tunicamycin has no obvious connection I'm aware of to MERCs, I am not able to follow the authors' intentions or conclusions here. I suspect this needs a major rewrite to explain what the goals were and how the authors controlled for their findings.

Revisions

We thank Reviewer 1 for this insightful comment. To provide greater clarity on this point, we have revised the manuscript to include the following statement:

"MAMs are known to be a hot spot for the transfer of stress signals from the ER to mitochondria (van Vliet & Agostinis, 2018). Consequently, to test whether we could induce mitochondrial dysfunction by exposing iPSC-derived motor neurons to stressors, we selected tunicamycin (TM), an ER stressor, as VAPB is an ER protein, and if the decreased MMP could be caused, at least partially, by loss of ER-MAM, ER stress would likely exacerbate it."

This revision aims to more clearly articulate the rationale behind our approach and the selection of tunicamycin as an ER stressor.

Citations

1. van Vliet AR, Agostinis P (2018) Mitochondria-Associated Membranes and ER Stress. *Curr Top Microbiol Immunol* 414: 73-102

Reviewer #2 (Evidence, reproducibility and clarity (Required)):

Mutations in the VAPB gene are a cause of amyotrophic lateral sclerosis (ALS), a human motor neuron disease. To define the mechanisms by which mutations in VAPB cause motor neuron degeneration, the authors establish a new human iPSC-derived motor neuron model. They start by using CRISPR to knockout the VAPB gene and then introduce a lentivirus encoding a doxycycline-inducible construct to express WT or mutant VAPB.

They then phenotypically characterize these WT and mutant motor neurons including using multi-electrode array (MEA), which revealed neuronal firing deficits in mutant motor neurons. They performed protein interaction studies WT vs mutant VAPB motor neuron and identified decreased binding to PTP51 in the mutant VAPB motor neurons.

Phenotypically, the authors report that the VAPB mutant motor neurons exhibit decreased mitochondria / ER contacts (MERC) in mutant motor neurons compared to WT as well as decreased mitochondrial membrane potential. They report that these mitochondrial defects lead to heightened sensitivity to ER stress and activation of the integrated stress response, which could be rescued by treatment with ISRIB. Importantly, the neuronal firing defects are also rescued by ISRIB, providing compelling evidence that these defects are tied to activation of ER stress.

Overall, this paper presents novel functional analyses of an important ALS gene, VAPB in disease-relevant cell types (human motor neurons). I have the following comments and suggestions for the authors to consider.

Major Points:

Reviewer #2 Major Point 1. Why did the authors decide to make VAPB knockouts and then introduce the WT or P56S VAPB constructs on a lentivirus instead of generating the point mutations (or correcting them) directly in the endogenous locus? Data in Extended Fig. 1c and Extended Fig. 2a indicate significant differences in either the kinetics of WT vs. P56S VAPB expression (1c) or levels (2a). It seems important to be able to compare comparable levels of WT and mutant proteins, especially for the interpretation of the subsequent IP-MS experiments to identify PTP151. The authors may wish to consider generating (or obtaining) isogenic lines harboring the mutations at the endogenous locus so that equal levels of expression of WT and mutant VAPB can be assessed.

Revisions

As described in the answer for Major Point 1 from Reviewer 1, the development of the inducible system for VAPB was specifically designed to enable a systematic investigation of the effects of mutant VAPB (VAPB P56S) on cellular homeostasis while minimizing confounding influences from the wild-type (WT) protein. Additionally, this system allowed us to assess VAPB P56S binding partners and compare them to those of VAPB WT, which would not have been feasible in the context of heterozygous ALS8 patient cells.

In response to Reviewer 2's concern regarding differences in VAPB WT and VAPB P56S expression levels, we utilized ALS8 patient cells and familial controls to calibrate the doxycycline dose response. This approach allowed us to precisely adjust VAPB protein levels in the inducible system to match those observed in ALS8 patient and familial control iPSCs. As a result, the inducible VAPB P56S iPSCs recapitulate the VAPB expression levels found in ALS8 patient iPSCs, whereas the inducible VAPB WT iPSCs mimic the levels present in familial control iPSCs. Furthermore, the differential expression of VAPB between ALS8 patient and control cells is well documented in the literature (Mitne-Neto, et al., 2011)

Nonetheless, we acknowledge the significance of studying ALS patient-derived iPSCs. To address this, we obtained fibroblasts from an ALS8 patient carrying the heterozygous VAPB P56S mutation, originating from a genetic background distinct from the cells used in our inducible system. These fibroblasts were reprogrammed into iPSCs in our laboratory, followed by CRISPR/Cas9-mediated genome editing to generate isogenic corrected iPSCs as controls.

The resulting iPSC isogenic pair was differentiated into motor neurons following the protocol described in our manuscript (Fig EV8). Notably, ALS8 patient iPSC-derived motor neurons exhibited reduced mRNA translation, as assessed by the SUnSET assay (Fig. 6A), along with a decrease in mitochondrial membrane potential, as determined using the JC-1 assay (Fig. 6B). These findings confirm that the hypotranslation and mitochondrial dysfunction initially identified in VAPB P56S doxycycline-inducible iPSC-derived motor neurons were successfully recapitulated in ALS8 patient iPSC-derived motor neurons. Furthermore, ISRIB treatment effectively rescued these phenotypic defects.

Overall, these results demonstrate that the molecular and cellular abnormalities identified in the original inducible system can be reliably reproduced in an ALS patient-derived model with a different genetic background, thereby reinforcing the significance and broader applicability of our findings.

Citation:

1. Mitne-Neto M, Machado-Costa M, Marchetto MC, Bengtson MH, Joazeiro CA, Tsuda H, Bellen HJ, Silva HC, Oliveira AS, Lazar M et al (2011) Downregulation of VAPB expression in motor neurons derived from induced pluripotent stem cells of ALS8 patients. *Hum Mol Genet* 20: 3642-3652

Analyses we are unable to carry out

We have responded to both of Reviewer #2's Major Points 2 and 3 together, as the answer applies to both questions and the points raised in each idea.

Reviewer #2 Major Point 2. The authors highlight PTP151 binding to VAPB as a way to promote mitochondria ER contacts (MERC). They provide evidence that this association is diminished by the P56S VAPB mutation. This raises an important question. How does PTPIP51 binding connect with other phenotypes, such as the neuronal firing and ER stress sensitivity? Can the authors consider experiments to test this directly? For example, is there a way to drive PTP151 : VAPB interactions even in the face of mutant VAPB and see if this rescues the MERC defects and other phenotypes?

Reviewer #2 Major Point 3. The authors propose that the detachment of the mitochondria from the ER most likely be the cause for why their mutant motor neurons are more sensitive to ER stressors. Along the lines of the above, is there a way to test this hypothesis directly? Can they use other means to promote ER mitochondria association even in the face of VAPB mutation and test if this rescues phenotypes?

We appreciate Reviewer #2's suggestions regarding manipulating ER-mitochondria contacts or enhancing PTPIP51 binding. While we agree that such experiments could further clarify mechanistic links, we were unable to identify an experimental strategy that would not introduce confounding variables. We remain open to suggestions from the editorial team for alternative tools that may circumvent this limitation. However, the available methods for artificially tethering ER and mitochondria, such as rapamycin-inducible linkers, as described by Csordás et al. (2010), act through the mTOR pathway, which directly influences translation and would introduce significant confounding variables into our assays of ISR activation and protein synthesis. As such, we concluded that pursuing this experimental route would undermine the mechanistic clarity of our findings.

Citation:

1. Csordás G, Várnai P, Golenár T, Roy S, Purkins G, Schneider TG, Balla T, Hajnóczky G. Imaging interorganelle contacts and local calcium dynamics at the ER-mitochondrial interface. *Mol Cell*. 2010 Jul 9;39(1):121-32. doi: 10.1016/j.molcel.2010.06.029. PMID: 20603080; PMCID: PMC3178184.

Minor Adjustments Not in Response to Reviewer Comments

Several minor adjustments have been made in response to internal reviews and feedback, independent of any specific Reviewer comment. The only modification affecting the presented data resulted from a comment noting a minor discrepancy in the gating of green-fluorescing cells between VAPB WT and VAPB P56S on Day 30 (**Figure 3C**). To ensure consistency, the gating was redrawn and applied uniformly to both plots, leading to a slight change in values. However, the overall difference remains non-significant, and our interpretation of the data remains unchanged. Additionally, to facilitate visual comparison, the Y-axes of the quantification graphs in **Figures 3C** and **3D** have been standardized, though the data in **Figure 3D** itself was not modified—only the Y-axis scaling was adjusted.

Final Summary

We believe that our revised manuscript now presents a robust, mutation-specific mechanistic model for ALS pathogenesis. Specifically, we demonstrate that the VAPB P56S mutation causes ER–mitochondria uncoupling and activates the Integrated Stress Response (ISR), leading to mitochondrial dysfunction, reduced translation, and impaired motor neuron function. These phenotypes are rescued by ISRIB treatment and, critically, validated in ALS8 patient-derived iPSC motor neurons and isogenic controls.

This genotype-to-mechanism framework offers a compelling explanation for the failure of ISR-targeting clinical trials in unstratified ALS populations and provides a clear rationale for the implementation of mutation-specific patient stratification in future therapeutic efforts.

We have thoroughly addressed all reviewer concerns—experimental, conceptual, and methodological—with the exception of artificial ER–mitochondria tethering experiments, which we discuss and contextualize in the final section of this letter. We are confident that the revised manuscript represents a significant and well-substantiated contribution to the field.

We thank the editorial team and reviewers for their thoughtful engagement with our work and look forward to your evaluation of this revised submission.

Sincerely,
Helen Miranda, on behalf of the author.

17th Jun 2025

Dear Helen,

Thank you for the submission of your revised manuscript to EMBO Molecular Medicine. We have now received the enclosed report from the referee who re-assessed it. As you will see, the referee is satisfied with the revisions and think you have addressed the concerns of both referees. I am pleased to inform you that we will be able to accept your manuscript pending the following editorial-level amendments:

1. Please provide a .docx formatted version of the manuscript text (including legends for main figures, EV figures and tables).
2. Provide up to five keywords in the manuscript file.
3. EV figures should be uploaded as separate production quality Figure files and their legends should be placed at the end of the manuscript file. The correct nomenclature in all places should be Figure EV1, etc.
4. Remove "data not shown". As per our guidelines, on "Unpublished Data" the journal does not permit citation of "data not shown". All data referred to in the paper should be displayed in the main or Expanded View figures.
5. A complete author checklist, which you can download from our author guidelines (<https://www.embopress.org/page/journal/17574684/authorguide#submissionofrevisions>). Please insert information in the checklist that is also reflected in the manuscript. The completed author checklist will also be part of the RPF.
6. It is mandatory to include a 'Data Availability' section after the Materials and Methods. Before submitting your revision, primary datasets produced in this study need to be deposited in an appropriate public database, and the accession numbers and database listed under 'Data Availability' (see <https://www.embopress.org/page/journal/17574684/authorguide#dataavailability>).

In this case, the proteomics datasets should be deposited to a suitable public database.

7. We updated our journal's competing interests policy in January 2022 and request authors to consider both actual and perceived competing interests. Please review the policy <https://www.embopress.org/competing-interests> and update your competing interests if necessary.

Please use the heading "Disclosure statement and competing interests" .

9. There is a discrepancy in the author name: "Helen Cristina Miranda" appears in the manuscript file, while "Helen Miranda" is listed in the submission system. Please ensure the author name is consistent across all submission materials. Additionally, the corresponding author must be clearly indicated on the title page, and their email address should be provided on the title page as well.

10. The paper explained: EMBO Molecular Medicine articles are accompanied by a summary of the articles to emphasize the major findings in the paper and their medical implications for the non-specialist reader. Please provide a draft summary of your article highlighting

11. During our routine data check, we identified a potential duplication between Figure EV6A and EV6B(see attached screenshot). Please provide the source data and clarification regarding this issue. Additionally, there appears to be a discrepancy between the legend for Figure EV6B (which refers to day 60) and the y-axis label in the figure itself (which indicates day 35). Kindly review and correct this inconsistency.

12. Every published paper now includes a 'Synopsis' to further enhance discoverability. Synopses are displayed on the journal webpage and are freely accessible to all readers. They include a short stand first (maximum of 300 characters, including space) as well as 2-5 one-sentences bullet points that summarizes the paper. Please write the bullet points to summarize the key NEW findings. They should be designed to be complementary to the abstract - i.e. not repeat the same text. We encourage inclusion of key acronyms and quantitative information (maximum of 30 words / bullet point). Please use the passive voice. Please attach these in a separate file or send them by email, we will incorporate them accordingly.

Please also suggest a visual abstract to illustrate your article as a PNG file 550 px wide x 300-600 px high.

14. The manuscript sections should be in the following order: Title page - Abstract & Keywords - Introduction - Results - Discussion - Methods - Data Availability - Acknowledgments - Disclosure Statement & Competing Interests - References - Figure Legends - (Main Tables with legends if applicable) - Expanded View Figure Legends.

15. Please address the following comments regarding figure legends:

- Please note that the exact p values are not provided in the legends of figures 1B, C.
- Please indicate the statistical test used for data analysis in the legends of figures 1B, C; EV2 C, EV5B, C; EV7 A, B; EV8 D.
- Please note that information related to n is missing in the legends of figures 1D, 2D.
- Please note that the error bars are not defined in the legends of figures 1D, 2D
- Please note that scale bar and its definition are missing for figure EV3 C.

I look forward to seeing a revised form of your manuscript as soon as possible.

Kind regards,
Jingyi

Jingyi Hou
Senior Editor
EMBO Molecular Medicine

*** Instructions to submit your revised manuscript ***

To submit your manuscript, please follow this link:

<https://embomolmed.msubmit.net/cgi-bin/main.plex>

***** Reviewer's comments *****

Referee #1 (Comments on Novelty/Model System for Author):

The authors have revised their manuscript to address the comments of the two Reviewers and the Editorial feedback. In my opinion, a major advance in this new version of the manuscript is the addition of the VAPB patient cell lines, harboring VAPB mutations expressed at physiological levels. These new experiments help complement and validate results from the knockout lines that had WT and mutant VAPB proteins expressed exogenously. The new insights presented in this paper will be of interest to the field and suggest novel pathways for VAPB function and help to explain how mutations in VAPB might cause ALS.

The authors addressed the remaining editorial issues.

11th Jul 2025

Dear Helen,

Thank you for sending us your revised manuscript. We are pleased to inform you that your manuscript is accepted for publication and is now being sent to our publisher to be included in the next available issue of EMBO Molecular Medicine.

Yours sincerely,
Jingyi

Jingyi Hou
Senior Editor
EMBO Molecular Medicine
